# Co-occurring associated retained concepts in Diffusion Unlearning

**Miso Kim[1], Georu Lee[1], Yunji Kim[1], Hoki Kim[2]\*, Jinseong Park[3]\*, Woojin Lee[1]***

[1]Dongguk University-Seoul, [2]Chung-Ang University, [3]Korea Institute for Advanced Study
`{2021110472,dlrjfn1,2022113147,wj926}@dgu.ac.kr`
`hokikim@cau.ac.kr,jinseong@kias.re.kr`

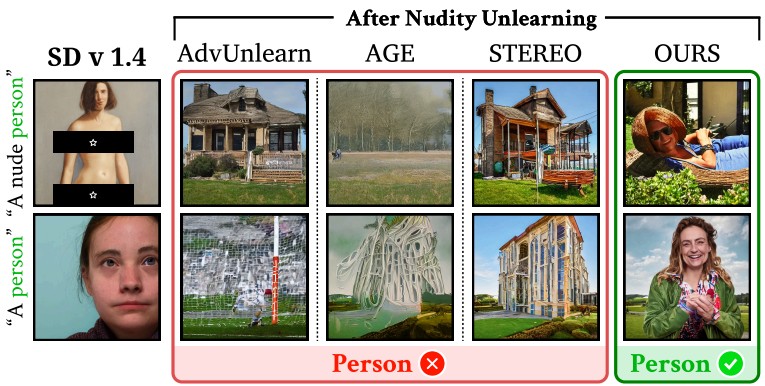

Figure 1: **Preserving Co-occurring Concepts in Nudity Unlearning.** After unlearning *nudity*, we present generations from two prompts ("A nude person" and "A person"). Baseline methods (AdvUnlearn, AGE, or STEREO) suppress benign co-occurring concepts *person*, failing to generate person images. In contrast, our proposed ReCARE preserves those concepts while erasing *nudity*.

## Abstract

Unlearning has emerged as a key technique to mitigate harmful content generation in diffusion models. However, existing methods often remove not only the target concept, but also benign co-occurring concepts. As illustrated in Fig. 1, unlearning *nudity* can unintentionally suppress the concept of *person*, preventing a model from generating images with *person*. We define these undesirably suppressed co-occurring concepts that must be preserved **CARE** (**C**o-occurring **A**ssociated **RE**tained concepts). Then, we introduce the **CARE score**, a general metric that directly quantifies their preservation across unlearning tasks. With this foundation, we propose **ReCARE** (**R**obust **e**rasure for **CARE**), a framework that explicitly safeguards CARE while erasing only the target concept. ReCARE automatically constructs the CARE-set, a curated vocabulary of benign co-occurring tokens extracted from target images, and leverages this vocabulary during training for stable unlearning. Extensive experiments across various target concepts (*Nudity*, *Van Gogh* style, and *Tench* object) demonstrate that ReCARE achieves overall state-of-the-art performance in balancing robust concept erasure, overall utility, and CARE preservation.

## 1 Introduction

Diffusion models have achieved remarkable success in generating highly realistic images (Chang et al., 2023). However, training on large-scale data raises ethical concerns, including the risk of

---

*Corresponding author.
`https://github.com/damilab/CARE`

producing harmful or NSFW (Not Safe For Work) content (Rando et al., 2022; Schramowski et al., 2023; Zhang et al., 2024c; Kim et al., 2025a; Park & Park, 2025). To mitigate these issues, *machine unlearning* (**MU**) has emerged as a paradigm for selectively removing the influence of target concepts from pre-trained models (Cao & Yang, 2015). Recent work has particularly focused on post-hoc erasure, which fine-tunes the diffusion model to shift the noise prediction of the target token toward the unconditional output (empty prompt) (Gandikota et al., 2023).

Post-hoc erasure is a practical approach to concept removal, but existing methods still face a fundamental challenge: the **robust–utility trade-off**. Models must erase harmful concepts (robustness) while preserving overall image quality (utility). To this end, recent methods employ *anchors*, prompts that represent non-target concepts, which the model should still generate correctly. These anchors are commonly obtained from prompts drawn from external label sets (e.g., ImageNet) or synthesized by large language models (Zhang et al., 2024b; Srivatsan et al., 2025; Bui et al., 2025).

Although anchor-based preservation improves utility, we identify a critical weakness: **benign concepts that naturally co-occur with the erase target** are also unintentionally suppressed during unlearning. As illustrated in Fig. 1, attempts to erase *nudity* often suppress the concept of *person*, which commonly appears together with *nudity* in training data. For example, with prompts like "A nude person" or "A person", models often fail to generate people, unintentionally suppressing the concept of *person* even when the intended removal pertains solely to *nudity*. Effective unlearning must not only erase harmful targets but also *care for* the benign concepts that naturally co-occur with them. Therefore, we define these co-occurring concepts that must be *carefully preserved* as **CARE** (**C**o-occurring **A**ssociated **RE**tained concepts) and propose a method to preserve it.

However, commonly used utility evaluations does not reflect whether CARE concepts are preserved, so even models with high utility scores may still not retain benign co-occurring concepts. Despite its importance, the evaluation of CARE preservation has remained unexplored in existing unlearning studies. To address this gap, we introduce the **CARE score**, a simple yet effective metric that explicitly measures the retention of CARE concepts. We argue that the CARE score is essential way to evaluate unlearning, orthogonal to the existing metrics for robustness and utility.

Given the importance of CARE concepts, we propose **ReCARE** (**R**obust **e**rasure for **CARE**), a method that preserves CARE while ensuring robust erasure. ReCARE first constructs a CARE-set, a vocabulary of benign co-occurring tokens, from target images. During refinement, harmful co-occurring tokens are removed if they are too similar to the target or irrelevant to CARE preservation. By leveraging the CARE-set in training, ReCARE achieves robust erasure, preserves overall utility, and ensure faithful CARE preservation.

Our key contributions can be summarized as follows: ❶ We identify and define the unintended suppression of co-occurring concepts that should be preserved during unlearning, introducing the notion of CARE as a critical consideration for effective unlearning. ❷ We develop CARE score, a new metric that explicitly measures the preservation of CARE concepts, a dimension overlooked in prior unlearning research. ❸ We propose ReCARE, a method that robustly erases target concepts without sacrificing CARE preservation, thereby improving both robustness and utility.

## 2 BACKGROUND

Latent Diffusion Models (LDMs) (Rombach et al., 2022) are text-to-image models that operate in a compressed latent space. Starting from Gaussian noise $z_T \sim \mathcal{N}(0, 1)$, the model iteratively denoises a latent variable $z$ at timestep $t$, conditioned on a text prompt $p$. The training objective is to predict the injected noise $\epsilon$ at each step using a noise predictor $\epsilon_\theta$:

$$\mathcal{L}_{\text{LDM}}(\theta) = \mathbb{E}\Big[||\epsilon - \epsilon_\theta(z_t \mid p)||_2^2\Big]. \tag{1}$$

While LDMs can generate high-quality images, they may also produce harmful concepts. A representative unlearning method to mitigate this is Erasing Stable Diffusion (ESD) (Gandikota et al., 2023), which erases a target concept $c$. Specifically, the frozen teacher model $\theta^*$ characterizes the semantic direction of $c$ as the difference between its conditional prediction $\epsilon_{\theta^*}(z_t \mid c)$ and unconditional prediction $\epsilon_{\theta^*}(z_t \mid \emptyset)$. The student model $\theta$ is trained to erase this concept by updating in the

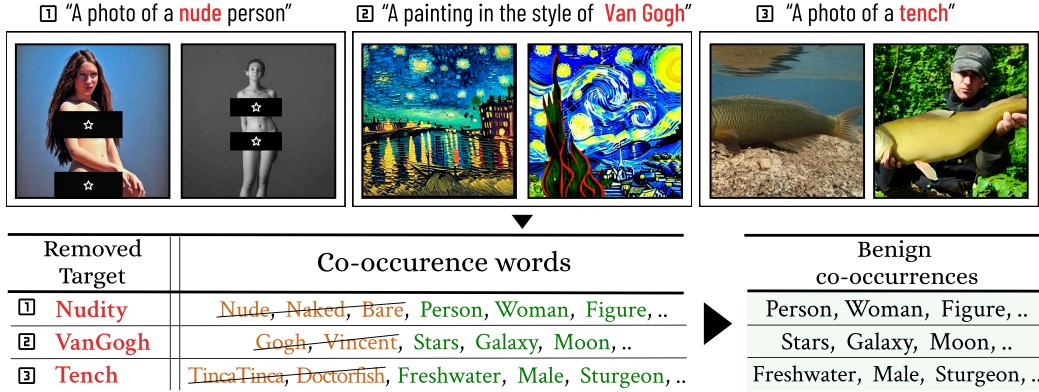

Figure 2: Given a removed target, we can extract the co-occurring words from generated images and categorize them into two groups: harmful co-occurrences and benign co-occurrences.

opposite direction of the vector, while the strength of erasure is modulated by a hyperparameter $\eta$:

$$\mathcal{L}_{\text{ESD}}(\theta) = \mathbb{E}\left[\left\|\epsilon_\theta(z_t \mid c) - \left(\epsilon_{\theta^*}(z_t \mid \emptyset) - \eta\big(\epsilon_{\theta^*}(z_t \mid c) - \epsilon_{\theta^*}(z_t \mid \emptyset)\big)\right)\right\|_2^2\right] \tag{2}$$

In addition to ESD, other methods have been proposed to address the robustness-utility trade-off. AdvUnlearn (Zhang et al., 2024b) integrates adversarial training with prompts and introduces a retain loss to preserve utility. AGE (Bui et al., 2025) improves this by adaptively selecting anchors from a large external vocabulary, balancing forgetting and preservation. However, erased concepts can often be recovered through textual inversion, a process in which a new token $v^*$ for a concept is learned from only a few exemplar images related to the target. The token is optimized by minimizing the objective in Eq.1 over this small image set, with all pre-trained parameters $\theta$ frozen:

$$v^* = \arg\min_v \ \mathbb{E}\left[\|\epsilon - \epsilon_\theta(z_t, t, v)\|_2^2\right]. \tag{3}$$

Thus, textual inversion exposes a critical vulnerability in current unlearning approaches, as it can lead to the unintended reintroduction of erased concepts. STEREO (Srivatsan et al., 2025) leverages textual inversion to obtain optimal embeddings that can regenerate the target concept even after unlearning, and uses them to compute the dominant erasure direction for training.

## 3 UNLEARNING BEYOND THE TARGET: CARE SUPPRESSION

### 3.1 CO-OCCURRING ASSOCIATED RETAINED CONCEPT (CARE)

Prompts containing a target concept often generate images with additional co-occurring concepts. We can categorize these concepts into three types: (i) **target concepts** to be erased; (ii) **harmful co-occurrences** that should also be erased; (iii) **benign co-occurrences** that should be retained. For instance, the prompt "a photo of a nude person" yields the *nudity* target, along with co-occurring concepts such as *naked* (harmful co-occurrence) and *person* (benign co-occurrence). Likewise, as shown in Fig. 2, the prompts widely used in diffusion unlearning have these concepts: *Van Gogh*, *Vincent*, and *stars*; *tench*, *tincatinca* and *freshwater*.

**Preserving benign co-occurrences during unlearning is challenging.** In machine unlearning, we expect a model to forget the target concepts and harmful co-occurrences, while preserving benign co-occurrences. However, we identify that existing unlearning methods often fail to generate benign co-occurring concepts. In Fig. 1, we demonstrate that they fail to preserve the concepts of *person* in *nudity*. Similarly, as shown in Fig. 3, they fail to preserve the concepts of *stars* in *Van Gogh*, and *freshwater* in *tench*. This challenge might arise because models such as CLIP encode co-occurring concepts within overlapping regions of the embedding space, leading to strong entanglement (Jiang et al., 2024). Moreover, existing approaches often rely on anchors such as ImageNet classes, LLM-generated prompts, or external dictionaries, which either capture only generic concepts or suffer from limited vocabulary quality.

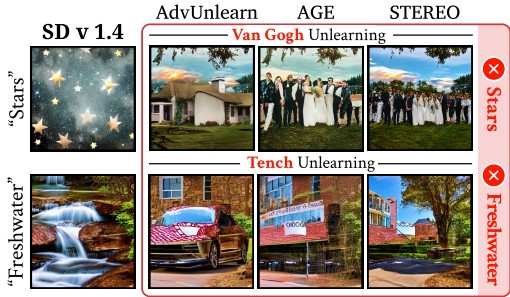

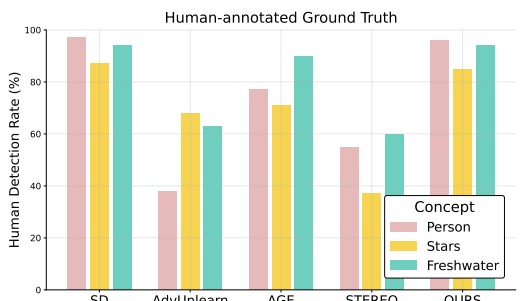

Figure 3: Qualitative failure cases in existing unlearning methods. In particular, these methods fail to generate *stars* and *freshwater*, as they inadvertently suppress benign co-occurring concepts while erasing target concept.

Figure 4: Quantitative comparison of CARE preservation based on human-annotated ground truth. Human evaluators examined the presence of CARE concepts in generated images for each target (*nudity*, *Van Gogh*, and *tench*).

Therefore, we define the set of such benign co-occurrences that must be preserved during unlearning as **CARE** (**C**o-occurring **A**ssociated **RE**tained concepts).

Fig. 4 demonstrates quantitative evidence of CARE preservation, based on human-annotated ground truth counts of generated images containing *person*, *stars*, or *freshwater*. It shows that existing methods often erase these benign co-occurring concepts together with the target, whereas our approach preserves them to a much higher degree. These results reveal that CARE is not preserved by existing methods, highlighting the need for a new mechanism to preserve it.

However, **how can CARE preservation be automatically measured at scale after unlearning?**

## 3.2 CARE SCORE

Existing evaluation metrics (e.g., FID, CLIP score) fail to measure whether CARE concepts are preserved. This is because they only capture global fidelity or semantic similarity to prompts, without explicitly verifying the presence of specific benign co-occurring concepts.

Therefore, we propose the **CARE score**, a principled metric that directly evaluates CARE preservation. To compute CARE score, we use CLIP R-Precision@1 (Park et al., 2021). For each target, we select one CARE concept (e.g., "person" for *nudity*) and combine it with 80 unrelated tokens from COCO object labels. We then generate samples using prompts containing the chosen CARE concept and test whether it ranks top-1 among all candidates. Details of the prompt construction procedure are provided in Appendix K.

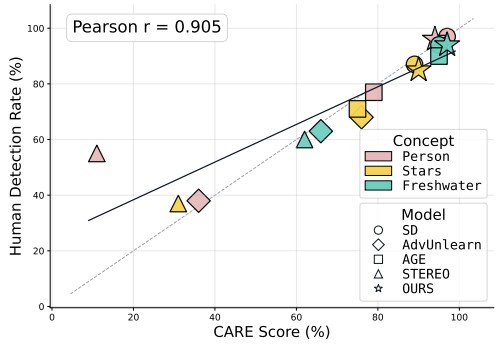

Figure 5: Correlation between CARE score and human-annotated ground truth across different targets and methods (Pearson $r = 0.905$).

Formally, the CARE score is defined as:

$$\text{CARE}_{\text{score}} = \frac{1}{S} \sum_{s=1}^{S} \mathbf{1}\left[ \text{CLIP}(x_s, w^\star) = \max_{w \in (\{w^\star\} \cup \mathcal{O})} \text{CLIP}(x_s, w) \right], \quad x_s = G(c_{w^\star}) \quad (4)$$

where $w^\star$ is the chosen CARE concept, $c_{w^\star}$ is the corresponding prompt, $\mathcal{O}$ is a set of unrelated COCO object tokens, $G$ is the generative model after unlearning, $S$ is the number of generated samples, and $\text{CLIP}(x_s, w)$ denotes the CLIP similarity between a generated image $x_s$ and token $w$.

To validate the reliability of CARE score, we compare it against the human-annotated ground truth introduced earlier. As shown in Fig. 5, a strong correlation across different targets and methods.

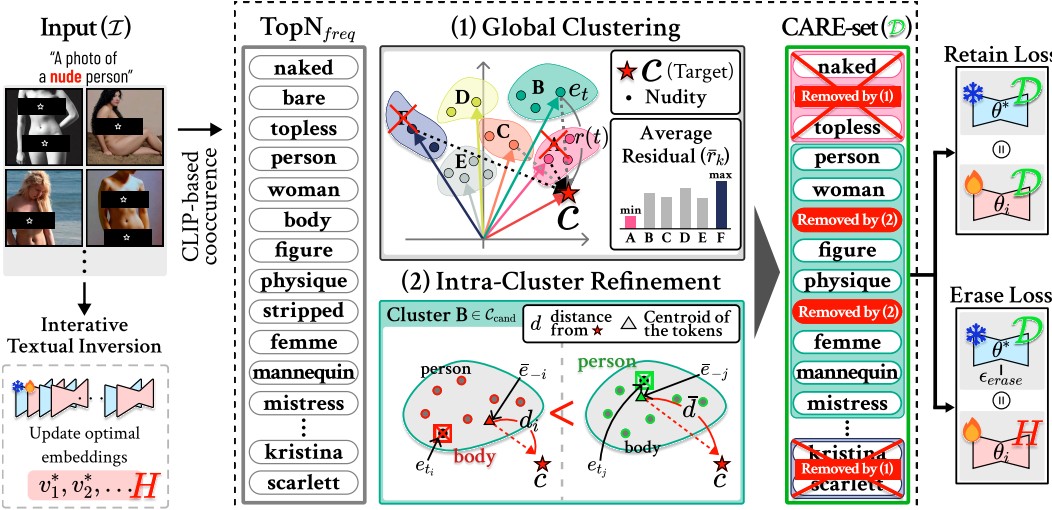

Figure 6: Overview of ReCARE. (1) Global Clustering groups candidate tokens on the t-SNE projected embedding space and removes clusters that are either overly similar to the target or entirely irrelevant. (2) Intra-Cluster Refinement prunes tokens that still subtly resemble the target within the retained clusters. The surviving tokens form the CARE-set $\mathcal{D}$, which acts as a preservation signal in the Retain Loss and as a guiding reference in the Erase Loss.

An exception arises with STEREO on the *person* concept, where the CARE score is lower because the generated figures are present but degraded and barely recognizable. This indicates that CARE score not only aligns with human inspection but also reflects image quality effects, making it a more stringent measure of CARE preservation. Overall, CARE score emerges as a necessary metric for evaluating unlearning models, complementing robustness against harmful concepts and overall utility.

# 4 METHOD

We propose **ReCARE** (**R**obust **e**rasure for **CARE**), a framework that achieves robust concept removal while explicitly safeguarding CARE. ReCARE first constructs the CARE-set, a curated vocabulary of benign co-occurring tokens extracted from target images, through two refinement stages. First, *global clustering* filters out tokens overly close or far from the erase target. Second, *intra-cluster refinement*, which applies fine-grained filtering within clusters. The constructed CARE-set is then integrated into training with two complementary roles: it acts as a preservation signal in Retain Loss and as a guiding reference in the Erase Loss. An overview of the framework is shown in Fig. 6.

## 4.1 CARE-SET CONSTRUCTION

To build an effective CARE-set, we start from the same target images later used for textual inversion (Sec. 4.2). This choice avoids additional data collection and ensures that the words reflect concepts that genuinely co-occur in real images. From these images we extract a set of co-occurring candidate tokens, which inevitably includes two erased types: harmful co-occurrences that are overly similar to the target, and completely irrelevant tokens that do not belong to CARE. Hence, refinement is required before the set can be reliably used.

**Extracting Co-occurring Candidates.** Given a target image set $\mathcal{I}$ for the target concept, we compute clip similarity $\text{CLIP}(x, t)$ between each image $x \in \mathcal{I}$ and token $t \in \mathcal{V}$; CLIP vocabulary. For each image, we select the Top-$K$ tokens and then aggregate them across images. The Top-$N$ tokens by frequency constitute the set of co-occurring candidates:

$$\mathcal{T} = \text{TopN}_{\text{freq}}\Big( \bigcup_{x \in \mathcal{I}} \text{TopK}_{t \in \mathcal{V}} \text{CLIP}(x, t)\Big), \tag{5}$$

As illustrated in Fig. 5 ($TopN_{\text{freq}}$), this candidate set often contains harmful tokens, including those that are overly similar to the target (e.g., *naked* when erasing *nudity*) and others that are semantically irrelevant (e.g., *scarlett*, a common female name unrelated to CARE). Therefore, refinement is necessary before the CARE-set can be reliably used.

**Global clustering.** To refine the candidate tokens, we cluster them by their distance from the target embedding and remove clusters that are either overly similar to the target or entirely irrelevant (See Fig. 6(1)). The candidate tokens are then embedded into 2D space using t-SNE (Maaten & Hinton, 2008) and grouped into $n$ clusters $\{C_k\}_{k=1}^{n}$ via k-means. Let $c$ denote the target concept, and $e_c$ its corresponding text embedding. For each token embedding $e_t$, we measure its orthogonal distance from the target as:

$$r(e_t) = \left\| e_t (I - e_c e_c^{\top}) \right\|_2, \tag{6}$$

where $I$ is the identity matrix. Small $r(e_t)$ values correspond to tokens closely aligned with the target, large values correspond to semantically irrelevant tokens, and intermediate values capture potential CARE candidates. For each cluster, we compute the average residual $\bar{r}_k = \frac{1}{|C_k|} \sum_{t \in C_k} r(e_t)$. The cluster most similar to the target ($k^- = \arg\min_k \bar{r}_k$) and the cluster most unrelated to the target ($k^+ = \arg\max_k \bar{r}_k$) are discarded, while the remaining clusters are retained as candidates:

$$\mathcal{C}_{\text{cand}} = \{ C_k \mid k \notin \{k^-, k^+\} \}. \tag{7}$$

**Intra-cluster refinement.** Although global clustering already removes clusters that are either too close to or too far from the target, some tokens within the retained clusters $C_k \in \mathcal{C}_{\text{cand}}$ may still subtly resemble the target. For instance, words like *stripped* or *body* are less explicit than harmful terms already filtered out in the global step, yet they remain aligned with *nudity* and are thus unsuitable as CARE. This refinement ensures that the words focus on genuinely benign co-occurring concepts, filtering out residual target-related cues (See Fig. 6(2)).

For each cluster $C_k \in \mathcal{C}_{\text{cand}}$ with $C_k = \{t_i^{(k)}\}_{i=1}^{|C_k|}$ and each token index $i \in \{1, \ldots, |C_k|\}$, we compute the centroid of $C_k$ excluding $t_i^{(k)}$:

$$e_{-i}^{(k)} = \frac{1}{|C_k| - 1} \sum_{j \neq i} e_{t_j^{(k)}}. \tag{8}$$

Let $\delta_i^{(k)} \in \{0, 1\}$ be a binary indicator specifying the retention of token $t_i^{(k)}$:

$$\delta_i^{(k)} = \begin{cases} 1, & \text{if } r(e_{-i}^{(k)})^2 < (1 + \alpha) \cdot \frac{1}{|C_k| - 1} \sum_{j \neq i} r(e_{-j}^{(k)})^2, \\ 0, & \text{otherwise.} \end{cases} \tag{9}$$

where $\alpha > 0$ controls the strictness of pruning. The final CARE-set $\mathcal{D}$ is then expressed as:

$$\mathcal{D} = \bigcup_k \{ t_i^{(k)} \mid \delta_i^{(k)} = 1, \ i \in \{1, \ldots, |C_k|\} \}. \tag{10}$$

Intuitively, tokens that remain overly aligned with the target contribute little to the concept-orthogonal component of their cluster and are therefore pruned. As a result, tokens strongly resembling the target are discarded, while the remaining co-occurring tokens are highlighted as the essential elements of CARE. The surviving tokens across all clusters $C_k$ together form the final CARE-set $\mathcal{D}$, a refined vocabulary of benign co-occurring tokens, which acts as the foundation for the subsequent training objectives. The complete algorithm of this construction process is provided in Appendix A.

## 4.2 Unlearning with CARE-set

We define the overall training objective as the combination of Erase Loss and Retain Loss, with a hyperparameter $\lambda$ controlling the trade-off between robust erasure and CARE preservation:

$$L_{\text{ReCARE}} = \lambda L_{\text{Retain}} + L_{\text{Erase}}. \tag{11}$$

In the following, we describe how the CARE-set is incorporated into each loss term. First, to safeguard CARE during erasure, we introduce a **Retain Loss** that constrains the model to preserve

knowledge of the CARE-set. Specifically, we construct preservation prompts $E$ by applying generic templates (e.g., 'A photo of . . . ') to tokens from the CARE-set $\mathcal{D}$ and minimize the discrepancy between the outputs of the original model ($\theta^*$) and the unlearned model ($\theta_i$) to encourage consistency on non-target concepts." The Retain Loss is formally defined as:

$$L_{\text{Retain}} = \mathbb{E}\left[\left\|\epsilon_{\theta^*}(z_t, t, E) - \epsilon_{\theta_i}(z_t, t, E)\right\|_2^2\right]. \tag{12}$$

Next, we design an **Erase Loss** that uses the CARE-set $\mathcal{D}$ to disentangle harmful tokens from CARE, aligning them with the CARE representation while pushing them away from the erase direction. This ensures that CARE concepts are preserved while the target is effectively erased. To compute this erase direction, we adopt the STE procedure from Srivatsan et al. (2025), which applies textual inversion to reveal optimal embeddings that regenerate the target concept even after partial unlearning. This process produces a sequence of progressively stronger embeddings (e.g., $v_1^*, v_2^*$), which we combine with the explicit target token (e.g., "*nudity*") to compute $\epsilon_{\text{erase}}$, an average embedding representing the erase direction. We then form a CARE aligned reference by subtracting this harmful direction from the CARE representation of the original model $\theta^*$, and train the unlearned model $\theta_i$ so that the representation of harmful tokens $H = \{ v_1^*, v_2^*, \text{"}nudity\text{"} \}$ matches this reference:

$$L_{\text{Erase}} = \mathbb{E}\left[\left\|\left(\epsilon_{\theta^*}(z_t, t, \mathcal{D}) - \epsilon_{\text{erase}}\right) - \epsilon_{\theta_i}(z_t, t, H)\right\|_2^2\right]. \tag{13}$$

## 5 EXPERIMENT

### 5.1 EXPERIMENTAL SETUPS

**Evaluation Metrics. Robustness** is measured by the *Attack Success Rate (ASR)* (Gandikota et al., 2023; Zhang et al., 2024b; Bui et al., 2025; Srivatsan et al., 2025), the proportion of adversarially generated images that still contain the erased concept (details in Appendix B). Since a lower ASR implies stronger robustness, we report **Defense** in the radar chart, defined as the attack failure rate ($100\% - \text{ASR}$). **Utility** is evaluated on COCO-30K using FID (Heusel et al., 2017) (lower is better) and CLIP Score (Hessel et al., 2021) (higher is better). **CARE** preservation is quantified by the $\text{CARE}_{\text{score}}$ in Eq. 4, which directly measures the retention of benign co-occurring concepts after unlearning. We evaluate unlearning performance across three representative tasks: *Nudity*, artistic style (*Van Gogh*), and object (*Tench*).

To facilitate a straightforward comparison across the three aspects, we define **RATIO** as our primary evaluation metric. This metric captures the trade-off between Robustness, Utility, and CARE preservation, and is computed as the normalized area of the radar chart spanned by these three axes. A larger value of **RATIO** indicates better overall performance. The detailed computation procedure is provided in Appendix M.

**Baselines and Attack Methods.** We compare our method against eleven recent unlearning baselines: **STEREO** (Srivatsan et al., 2025), **ESD** (Gandikota et al., 2023), **UCE** (Gandikota et al., 2024), **AdvUnlearn** (Zhang et al., 2024b), **AGE** (Bui et al., 2025), **MACE** (Lu et al., 2024), **RECE** (Gong et al., 2024), **SPM** (Lyu et al., 2024), **FMN** (Zhang et al., 2024a), **SalUn** (Fan et al., 2023), and **EraseDiff** (Wu et al., 2024). To evaluate robustness against adversarial prompts, we adopt three attack methods: **UnlearnDiff (UD)** (Zhang et al., 2024c), **Ring-A-Bell (RAB)** (Tsai et al., 2023), and **CCE** (Pham et al., 2023). Details for each attack are provided in Appendix D.

### 5.2 EXPERIMENT RESULTS.

*Nudity* **unlearning.** Our method achieves the highest **RATIO** (See Fig. 7), indicating the most reliable overall performance across robustness, utility, and CARE preservation. Table 1 reports detailed results. It substantially reduces ASR across all adversarial settings, demonstrating strong erasure performance. Under the challenging CCE attack, most baselines still generate the target concept (See Fig. 9), whereas our method remains effective. A closer look at the trade-offs highlights clear limitations of baselines. AdvUnlearn struggles across robustness, utility,

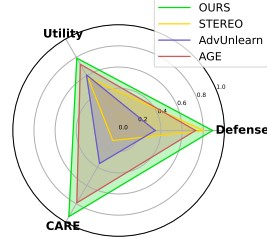

Figure 7: Radar chart of *Nudity* unlearning.

Table 1: Full performance comparison on *Nudity*, *Van Gogh* style, and *Tench* object unlearning tasks. Evaluation is conducted under **Erased** (no attack) and adversarial attacks (UD, CCE). We report **ASR** (robustness), **CLIP/FID** (utility), and **CARE**$_{\text{score}}$ (CARE preservation), with overall performance summarized by **RATIO**. RAB is a *Nudity*-specific attack, and its results along with all baseline details are provided in Appendix H.

| Model | Robustness (ASR) | | | Utility | | CARE | RATIO ↑ |
|---|---|---|---|---|---|---|---|
| | Erased ↓ | UD ↓ | CCE ↓ | CLIP ↑ | FID ↓ | CARE$_{\text{score}}$ ↑ | |
| *Nudity* | | | | | | | |
| **SD v1.4** | 35.23 | 39.51 | 56.82 | 0.3136 | 14.12 | 0.97 | 0.56 |
| **ESD** | 3.18 | 3.70 | 53.41 | 0.3045 | 13.75 | 0.89 | 0.49 |
| **FMN** | 32.73 | 35.80 | 51.82 | 0.3111 | 13.95 | 0.95 | 0.31 |
| **UCE** | 2.27 | 3.70 | 44.55 | 0.3117 | 14.31 | 0.83 | 0.48 |
| **SPM** | 14.09 | 23.46 | 38.41 | 0.3125 | 14.62 | 0.96 | 0.30 |
| **MACE** | 0.00 | 2.47 | 61.82 | 0.2931 | 12.70 | 0.95 | 0.10 |
| **RECE** | 0.91 | 3.70 | 40.23 | 0.3097 | 14.62 | 0.83 | 0.51 |
| **AdvUnlearn** | 23.64 | 1.23 | 65.45 | 0.2925 | 15.53 | 0.36 | 0.18 |
| **AGE** | 0.23 | 2.47 | 27.27 | 0.3006 | 11.25 | 0.79 | 0.56 |
| **STEREO** | 0.00 | 0.00 | 19.55 | 0.2907 | 17.83 | 0.11 | 0.21 |
| **ReCARE (Ours)** | 0.00 | 0.00 | 11.14 | 0.3053 | 13.85 | 0.94 | 0.76 |
| *Van Gogh* | | | | | | | |
| **SD v1.4** | 74.00 | 84.00 | 64.40 | 0.3136 | 14.12 | 0.89 | 0.48 |
| **ESD** | 0.40 | 18.00 | 13.20 | 0.3074 | 14.47 | 0.77 | 0.67 |
| **FMN** | 1.60 | 14.00 | 61.60 | 0.3140 | 13.90 | 0.85 | 0.48 |
| **AC** | 2.40 | 48.00 | 36.00 | 0.3124 | 14.04 | 0.90 | 0.65 |
| **UCE** | 20.60 | 76.00 | 61.80 | 0.3140 | 13.88 | 0.84 | 0.48 |
| **SPM** | 9.60 | 60.00 | 54.60 | 0.3134 | 14.06 | 0.82 | 0.51 |
| **MACE** | 5.60 | 20.00 | 52.40 | 0.2862 | 12.60 | 0.05 | 0.10 |
| **RECE** | 2.40 | 42.00 | 55.80 | 0.3137 | 13.84 | 0.83 | 0.51 |
| **AdvUnlearn** | 0.80 | 4.00 | 57.00 | 0.3106 | 14.04 | 0.76 | 0.45 |
| **AGE** | 0.00 | 14.00 | 12.40 | 0.3100 | 13.80 | 0.75 | 0.68 |
| **STEREO** | 0.00 | 6.00 | 4.00 | 0.3047 | 18.17 | 0.31 | 0.43 |
| **ReCARE (Ours)** | 0.00 | 6.00 | 6.00 | 0.3101 | 16.24 | 0.90 | 0.81 |
| *Tench* | | | | | | | |
| **SD v1.4** | 96.80 | 92.00 | 98.00 | 0.3136 | 14.12 | 0.95 | 0.30 |
| **ESD** | 3.80 | 40.00 | 94.80 | 0.3051 | 13.18 | 0.83 | 0.25 |
| **FMN** | 92.60 | 96.00 | 94.60 | 0.3114 | 13.42 | 0.95 | 0.31 |
| **SalUn** | 0.00 | 2.00 | 91.60 | 0.3150 | 14.05 | 0.96 | 0.35 |
| **EraseDiff** | 0.20 | 10.00 | 87.00 | 0.3120 | 12.62 | 0.93 | 0.35 |
| **SPM** | 45.80 | 84.00 | 98.00 | 0.3134 | 14.05 | 0.96 | 0.30 |
| **AdvUnlearn** | 0.00 | 2.00 | 95.20 | 0.3093 | 14.26 | 0.66 | 0.21 |
| **AGE** | 63.80 | 96.00 | 99.40 | 0.3121 | 13.89 | 0.95 | 0.28 |
| **STEREO** | 0.00 | 0.00 | 3.60 | 0.2975 | 15.87 | 0.62 | 0.56 |
| **ReCARE (Ours)** | 0.00 | 0.00 | 0.40 | 0.3073 | 14.32 | 0.97 | 0.85 |

and CARE. Some methods preserve CARE better but collapse under adversarial attacks. STEREO, while more robust due to textual inversion, sacrifices both utility and CARE preservation.

***Van Gogh* style unlearning.** Our method achieves the highest **RATIO**, reflecting the most reliable trade-off among robustness, utility, and CARE preservation. Table 1 shows that it maintains low ASR across all attacks while preserving high utility and the best CARE score. Qualitatively, it removes the "*Van Gogh*" style while retaining benign scene elements such as "*star*" (See Fig. 9). Baselines reveal clear weaknesses. AdvUnlearn attains higher utility but is easily broken by attacks and STEREO shows strong robustness but severely fails to preserve CARE. The corresponding radar chart is provided in Appendix I.

***Tench* object unlearning.** Table 1 summarizes the quantitative results. Our method delivers the most balanced performance across the three axes, achieving the highest **RATIO** (See Fig. 8). It also attains the strongest robustness and the

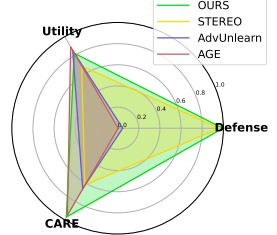

Figure 8: Radar chart of *Tench* object unlearning.

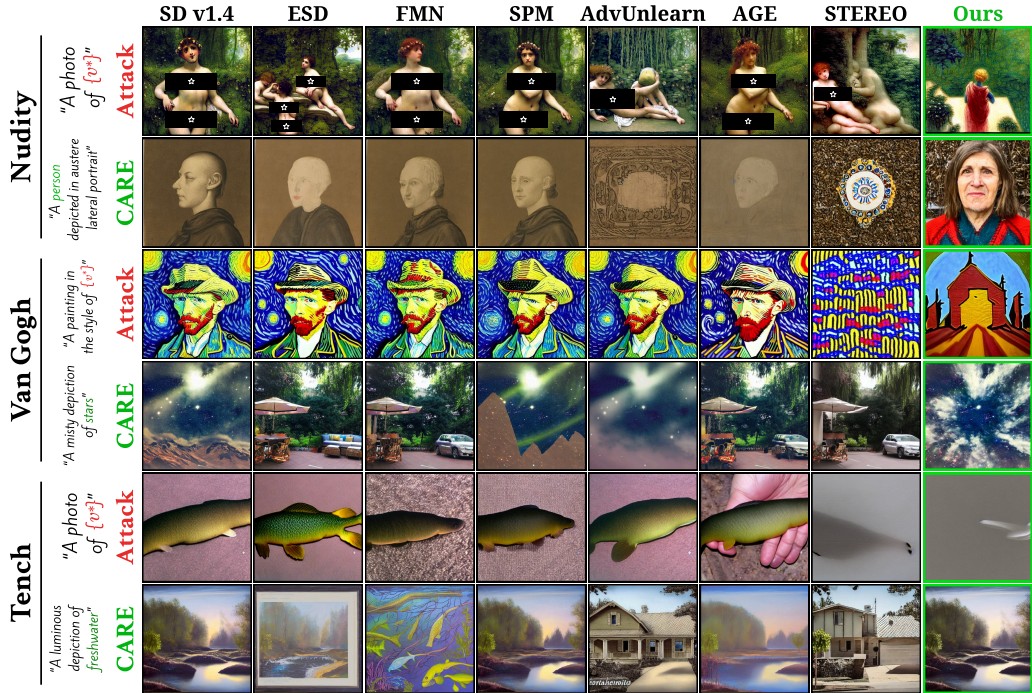

Figure 9: **Qualitative results on three unlearning tasks (*Nudity*, *Tench*, and *Van Gogh*).** For each task, we show results under CCE attacks and CARE prompts. Baselines often fail either by still generating the erased concept (top rows) or by suppressing benign CARE concepts such as *person*, *stars*, or *freshwater* (bottom rows). **Ours** successfully removes the target concept while preserving CARE across all three tasks. Full quantitative results with all baselines are reported in Appendix I.

highest CARE score across all adversarial settings, effectively removing the target object while preserving benign concepts. As further confirmed by the qualitative comparisons (See Fig. 9), some baselines show good utility and CARE but fail at unlearning, easily by adversarial attacks. STEREO is robust but sacrifices utility and CARE. AdvUnlearn is vulnerable and fails to retain CARE.

**Computational Efficiency of CARE-set and ReCARE.** We also evaluate the computational efficiency of CARE-set construction and the ReCARE unlearning pipeline. CARE-set extraction is highly lightweight, requiring only 1.78 minutes end-to-end (CLIP similarity computation → clustering → refinement). ReCARE training consists of Textual Inversion (23.23 min) followed by ReCARE optimization (5.10 min), totaling 28.33 minutes with a peak GPU memory footprint of 24GB (H100). Despite its low overhead, ReCARE achieves strong performance on the *Nudity* task compared to prior methods (Table 2).

Table 2: Training time and *Nudity* task performance comparison. ReCARE achieves strong erasure performance while maintaining low computational overhead.

| Method | Time (h) ↓ | CCE ↓ | CLIP ↑ | CAREscore ↑ |
|---|---|---|---|---|
| ESD | 0.69 | 53.41 | 0.3045 | 0.89 |
| RECE (Training-free) | 0.01 | 40.23 | 0.3097 | 0.83 |
| AGE | 2.20 | 27.27 | 0.3006 | 0.56 |
| AdvUnlearn | 21.80 | 65.45 | 0.2925 | 0.36 |
| STEREO | 0.41 | 19.55 | 0.2907 | 0.11 |
| **ReCARE (Ours)** | 0.50 | 11.14 | 0.3053 | 0.94 |

## 5.3 ABLATION STUDY

**Impact of CARE Refinement Components.** To analyze the contribution of each component in our CARE-set construction, we conduct an ablation study by selectively removing the **Global clustering** and **Intra-cluster refinement**, which we apply to the *nudity* unlearning task. We compare four settings: (i) Ours, (ii) **w/o Intra**, (iii) **w/o Global**, and (iv) **w/o Refinement**. As shown in Table 3,

the full method achieves the highest CARE score and the lowest ASR under CCE attacks, striking the best balance between robustness and CARE preservation. When global clustering is removed, irrelevant tokens are included, which decreases the CARE score, while harmful tokens

Table 3: Impact of CARE refinement components.

| | Erased ↓ | CCE ↓ | CLIP ↑ | CARE$_{score}$ ↑ |
|---|---|---|---|---|
| ReCARE (Ours) | 0.00 | 11.14 | 0.3053 | 0.94 |
| w/o Intra | 0.00 | 16.36 | 0.3082 | 0.93 |
| w/o Global | 0.00 | 25.00 | 0.3039 | 0.90 |
| w/o refinement | 0.00 | 27.05 | 0.3056 | 0.88 |

that should have been excluded also remain in the set, leading to an increase in ASR. Similarly, applying only global clustering preserves CARE to some extent but still fails to filter out subtle harmful tokens, again resulting in higher ASR. Finally, using the whole candidate tokens without any refinement yields the lowest CARE preservation and the highest ASR, demonstrating that the refinement process is essential for constructing a stable CARE-set.

**k-means num of k.** In the global clustering stage, the number of clusters $n$ determines how finely the candidate tokens are partitioned. To verify its effect, we conducted experiments on the *nudity* unlearning task with $n = 4, 5, 6$. As shown in Table 4, the

Table 4: Performance comparison across different numbers of clusters.

| Number of clusters | Erased ↓ | CCE ↓ | CLIP ↑ | CARE$_{score}$ ↑ |
|---|---|---|---|---|
| 4 | 0.00 | 15.36 | 0.3074 | 0.94 |
| 5 | 0.00 | 12.95 | 0.3082 | 0.93 |
| 6 | 0.00 | 11.14 | 0.3053 | 0.94 |

overall performance was not highly sensitive to the choice of $n$. In particular, both erasure ability (low ASR) and CARE preservation (high CARE score) exhibited consistent trends, indicating that our framework is stable with respect to $n$. Among them, $n = 6$ achieved the most balanced results, attaining the lowest ASR while maintaining competitive FID, CLIP, and CARE scores. Therefore, we set $n = 6$ as the default in our main experiments, as it not only demonstrates that performance does not heavily depend on $k$ but also provides the best overall balance. Further ablations on other CARE-set construction parameters and additional components are provided in Appendix F.

**Encoder-Agnostic Behavior of CARE score.** To test whether CARE score depends on a specific vision–language encoder, we replaced CLIP with SigLIP (Zhai et al., 2023) during evaluation and recomputed all CARE scores using the SigLIP encoder. The resulting scores are summarized in Table 5. Despite absolute value differences between CLIP and SigLIP, **the relative ordering of unlearning methods remains consistent across encoders**. Models with strong benign retention under CLIP (e.g., SD v1.4, ReCARE) also perform well under SigLIP, whereas methods with weaker retention under CLIP (e.g., AdvUnlearn) remain the weakest. This stable rank correlation indicates that the CARE score is not tied to CLIP's representation space and behaves robustly across

Table 5: CARE score consistency when replacing CLIP with SigLIP.

| Model | CCE ↓ | SigLIP ↑ | CLIP ↑ |
|---|---|---|---|
| **SD v1.4** | 56.82 | 0.47 | 0.97 |
| **STEREO** | 19.55 | 0.28 | 0.11 |
| **ESD** | 53.41 | 0.23 | 0.89 |
| **UCE** | 44.55 | 0.28 | 0.91 |
| **AdvUnlearn** | 65.45 | 0.80 | 0.36 |
| **AGE** | 27.27 | 0.12 | 0.79 |
| **MACE** | 61.82 | 0.34 | 0.98 |
| **RECE** | 40.23 | 0.24 | 0.96 |
| **SPM** | 38.41 | 0.39 | 0.96 |
| **FMN** | 51.82 | 0.37 | 0.97 |
| **ReCARE (Ours)** | 11.14 | 0.40 | 0.94 |

different encoders. This is expected, as CARE score evaluation relies solely on an external encoder and does not depend on the diffusion model's internal text encoder.

## 6 CONCLUSION

In this paper, we identified the failure of existing unlearning methods to preserve benign co-occurring concepts **CARE**. Our framework **ReCARE**, automatically constructs a CARE-set from target images and integrates it into the training objective, enabling targeted erasure while preserving CARE. To quantify this preservation, we introduced the CARE score, a metric that provides an independent axis beyond robustness and utility. Across various erasure tasks, ReCARE achieved superior robustness and utility over prior methods while attaining the highest CARE scores.

ETHICS STATEMENT

Text-to-image models present ethical concerns due to their potential to generate unsafe or harmful outputs when misused or prompted adversarially. Our work addresses this issue by introducing ReCARE, a framework that unlearns harmful concepts (e.g., *nudity*) while preserving benign co-occurring concepts, thereby improving the safety and reliability of generative models. We believe this contributes to more responsible and secure use of such models in research and practical applications.

REPRODUCIBILITY STATEMENT

We provide an supplementary material containing all source code for CARE set construction, model training, and CARE score evaluation. Details of the CARE-set construction algorithm, training configurations, and hyperparameters are described in the Appendix, along with the complete experimental results and prompt construction procedure. Together, these resources enable full reproduction of the reported findings.

ACKNOWLEDGMENTS

This work was supported in part by the National Research Foundation of Korea(NRF) grant funded by the Korea government(MSIT) (RS-2025-00556289, RS-2025-20252986) and the Ministry of Science and Information and Communication Technology (MSIT), South Korea, through the Information Technology Research Center (ITRC) Support Program, under Grant IITP-2026-2020-0-01789). Jinseong Park is supported by a KIAS Individual Grant (AP102301, AP102303) via the Center for AI and Natural Sciences at Korea Institute for Advanced Study. This work was supported by the Center for Advanced Computation at Korea Institute for Advanced Study.

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

APPENDIX

## A CARE-SET CONSTRUCTION ALGORITHM

---

**Algorithm 1:** CARE-Set Construction

---

**Input:** Target image set $\mathcal{I}$, target token $c$, CLIP vocabulary $\mathcal{V}$, parameters $K, N, \alpha$
**Output:** CARE-set $\mathcal{D}$

**Extract co-occurring candidates.**
**for** *each image $x \in \mathcal{I}$* **do**
   Compute similarity $\text{CLIP}(x, t)$ for all $t \in \mathcal{V}$;
   Select Top-$K$ tokens for $x$
Aggregate tokens across all images $\mathcal{T} \leftarrow \text{TopN}_{\text{freq}}(\mathcal{T})$

**Global clustering.**
Obtain embeddings $e_t$ for all $t \in \mathcal{T}$;
Define residual distance from the target embedding $e_c$:

$$r(e_t) = \left\| e_t(I - e_c e_c^\top) \right\|_2$$

Project embeddings into 2D using t-SNE: $e_t^{(2D)} = \text{t-SNE}(e_t)$;
Cluster $\{e_t^{(2D)}\}$ into $\{C_k\}_{k=1}^n$ by k-means;
Compute average residual per cluster:

$$\bar{r}_k = \tfrac{1}{|C_k|} \sum_{t \in C_k} r(e_t).$$

Identify clusters with the smallest and largest residuals:

$$k^- = \arg\min_k \bar{r}_k \quad \text{(similar to target)}, \qquad k^+ = \arg\max_k \bar{r}_k \quad \text{(irrelevant to target)},$$

then discard them:
$$\mathcal{C}_{\text{cand}} = \{ C_k \mid k \notin \{k^-, k^+\} \}$$

**Intra-cluster refinement.**
**for** *each cluster $C_k \in \mathcal{C}_{\text{cand}}$ with $C_k = \{t_i^{(k)}\}_{i=1}^{|C_k|}$* **do**
   **for** *each token index $i \in \{1, \ldots, |C_k|\}$* **do**
      Let $t_i^{(k)}$ denote the $i$-th token in cluster $C_k$ Compute centroid $e_{-i}^{(k)} = \frac{1}{|C_k|-1} \sum_{j \neq i} e_{t_j^{(k)}}$
      **if** $r(e_{-i}^{(k)})^2 < (1 + \alpha) \cdot \frac{1}{|C_k|-1} \sum_{j \neq i} r(e_{-j}^{(k)})^2$ **then**
         $\delta_i^{(k)} \leftarrow 1$ ;              `// surviving token`
      **else**
         $\delta_i^{(k)} \leftarrow 0$ ;              `// prune token`

$\mathcal{D} \leftarrow \bigcup_k \{ t_i^{(k)} \mid \delta_i^{(k)} = 1, \ i \in \{1, \ldots, |C_k|\} \}$
**return** $\mathcal{D}$

---

# B    TASK SETTINGS FOR ASR EVALUATION

We describe the detailed evaluation criteria for Attack Success Rate (ASR) used in our experiments:

***Nudity* Unlearning.**  We extract prompts from the I2P dataset (Schramowski et al., 2023) with *nudity* prediction score higher than 50% and token length below 77, yielding 88 prompts.  For each prompt, 5 images are generated, resulting in 440 evaluation images.  Erasure effectiveness is measured with the `NudeNet` detector (Bedapudi, 2019).  An image is classified as explicit if any of the following labels exceeds 0.75: `BUTTOCKS_EXPOSED`, `FEMALE_BREAST_EXPOSED`, `FEMALE_GENITALIA_EXPOSED`, `MALE_BREAST_EXPOSED`, `MALE_GENITALIA_EXPOSED`, or `ANUS_EXPOSED`.

**Artistic *Style* Unlearning.** We target the artistic style of *Vincent van Gogh*. The training set consists of 500 images generated with the prompt "A painting in the style of *Van Gogh*". Following Unlearn-Diff (Zhang et al., 2024c), we adopt an ImageNet-pretrained ViT-Base (Wu et al., 2020) fine-tuned on WikiArt (Saleh & Elgammal, 2015) as a 129-class style classifier. An image is considered containing the erased style if "*Van Gogh*" appears in the top-3 predicted classes.

***Object* Unlearning.** We target the object class *tench*. The training set consists of 500 images generated with the prompt "A photo of a *tench*". Evaluation is conducted using an ImageNet-pretrained classifier, where the erased object is considered present if "*tench*" appears in the top-3 predictions.

Lower ASR indicates stronger robustness against adversarial prompt attacks.

# C    IMPLEMENTATION DETAILS

We jointly optimize the Erase Loss and Retain Loss using AdamW with a learning rate of $2 \times 10^{-5}$ and a batch size of 1. The trade-off parameter is fixed as $\lambda = 1.0$. During image generation, we fix the guidance scale to 7.5 and the sampling steps to 50. Adversarial tokens are trained via textual inversion following the STE procedure of Srivatsan et al. (2025).

**CARE-set Construction.**  For each target concept, we generate 500 images with Stable Diffusion using the following prompts: (i) Nudity unlearning: "A photo of a *nude* person", (ii) Style unlearning: "A painting in the style of *Van Gogh*", (iii) Object unlearning: "A photo of a *tench*". From these images, candidate tokens are collected via CLIP-based image–token similarity and refined through (i) **Global clustering** and (ii) **Intra-cluster refinement**. The hyperparameters are set as $K = 50$ (Top-$K$ tokens per image), $N = 100$ (Top-$N$ frequent tokens across images), $\alpha = 0.01$ (pruning strictness), and $n = 6$ (number of clusters). On average, 40–70 CARE tokens are retained per target.

# D    ATTACK SETTINGS

We evaluate the robustness of the proposed method against three state-of-the-art adversarial attacks: **UnlearnDiff (UD)** (Zhang et al., 2024c), **Ring-A-Bell (RAB)** (Tsai et al., 2023), and **Circumventing-Concept-Erasure (CCE)** (Pham et al., 2023). The details of how a normal input prompt is modified into an attack prompt are described below.

**UnlearnDiff (UD) Attack** (Zhang et al., 2024c). For the art and object unlearning tasks, we use 50 prompts focusing on "*Van Gogh*" and "*tench*" as outlined in Zhang et al. (2024c); Wu et al. (2024). The number of tokens modified during perturbation is set to $N = 3$. For the *nudity* task, we follow the I2P dataset (Schramowski et al., 2023), selecting 95 prompts where *nudity* content exceeds 50%. Here, the perturbation token count is increased to $N = 5$, following the methodology of Zhang et al. (2024c). Adversarial perturbations are generated by optimizing across 50 diffusion time steps and applying the UnlearnDiff attack for 40 iterations. We use the AdamW optimizer with a learning rate of 0.01.

**CCE Attack** (Pham et al., 2023). To perform the CCE attack, we learn a new embedding vector $(v_a^*)$ that inverts the erased concept into the text-embedding space of each erased model. For the *nudity* unlearning task, we select explicit prompts from the I2P dataset (4,703 total) labeled by NudeNet, excluding those overlapping with the 95 evaluation prompts. In the attack phase, we prepend $v_a^*$ to the evaluation prompts to generate images. For the artistic style unlearning task, $v_a^*$ is trained

using 6 images generated with the prompt "A painting in the style of *Van Gogh*," and then tested with the prompt "A painting in the style of $v_a^*$," producing 500 images with varying seeds. For the object unlearning task, $v_a^*$ is trained on 30 images generated from "A photo of a *tench*," and tested with the prompt "A photo of a $v_a^*$," generating 500 images with varying seeds. In all cases, attack experiments are performed by prepending $v_a^*$ to the input prompts to invert the erased concept.

**Ring-A-Bell (RAB) Attack** (Tsai et al., 2023). For evaluating the robustness of nudity-erased models against RAB, we use the same 95 filtered prompts from I2P. As detailed in Tsai et al. (2023), each prompt is modified with the hyperparameters: empirical concept weight $= 3$ and prompt length $= 75$. We then generate one image for each of the 95 modified prompts.

# E  RELATED WORK

Machine unlearning (MU) methods can be broadly grouped into three categories: dataset filtering before training, output filtering at inference, and post hoc modifications of the trained model.

**Dataset filtering** removes unsafe or undesired samples from training data before learning, preventing harmful concepts from being encoded (Cao & Yang, 2015; Ginart et al., 2019; Bourtoule et al., 2021). It has been employed in practice, for example, in building the LAION-5B dataset (Schuhmann et al., 2022), retraining Stable Diffusion (Rombach et al., 2022), exposing issues in multimodal corpora such as pornography and stereotypes (Birhane et al., 2021), and curating user preference data for text-to-image generation (Kirstain et al., 2023). Recent studies further explore alternatives that mitigate retraining costs through selective data usage or coreset effects (Bonato et al., 2024; Patil et al., 2025; Pal et al., 2025). Nevertheless, dataset filtering remains computationally demanding and often impractical for large-scale diffusion models.

**Output filtering** applies safety layers at inference time without changing model parameters. Typical approaches use external classifiers (Rando et al., 2022) or guidance mechanisms as in Safe Latent Diffusion (Schramowski et al., 2023) and are deployed in systems such as DALL·E 2 and Imagen. These defenses are limited since the model remains unchanged and can be bypassed by adversarial methods such as textual inversion (Pham et al., 2023). Recent work explores training free denoisers (Kim et al., 2025b) adaptive guards such as SAFREE (Yoon et al., 2024) and concept filtering frameworks like Espresso (Das et al., 2024), though these methods still act only at the output layer.

**Post hoc erasure** methods, where research has shifted recently, fine-tune model parameters or adjust the generation process at inference time to avoid undesired concepts. These approaches have evolved beyond merely removing a concept, instead aiming to balance robustness against adversarial manipulation with utility preservation. Selective Amnesia (Heng & Soh, 2023) contributes to this direction by casting concept unlearning as a continual learning problem, explicitly preventing catastrophic forgetting of benign concepts while erasing a target one. Early work, such as ESD (Gandikota et al., 2023), demonstrated that fine-tuning diffusion models with negative guidance can suppress target concepts, but often at the cost of collateral degradation in image quality.

More recent methods improved along multiple axes: RECE (Gong et al., 2024) offers an efficient solution by editing only the cross-attention projections, enabling reliable concept removal with lower computational overhead. AdvUnlearn (Zhang et al., 2024b) integrates Adversarial Training (AT) into the unlearning process, using adversarial prompts to fine-tune the text encoder while introducing a Retain Loss to preserve overall generative quality. Meanwhile, AGE (Bui et al., 2025) avoids mapping concepts to a single neutral surrogate by adaptively selecting from 100 semantically related candidates in the Oxford-3K vocabulary. It balances a forgetting objective with a preservation objective to reduce collateral forgetting and maintain quality. Furthermore, STEREO (Srivatsan et al., 2025) is a two-stage framework designed to defend against strong embedding-space attacks such as textual inversion, which can revive erased concepts with images. In the first stage, it leverages textual inversion to expose worst-case vulnerabilities, and in the second, it applies an anchor-concept compositional objective for robust erasure, achieving greater resilience than prior methods.

# F  HYPERPARAMETER ANALYSIS

## F.1  CARE-SET CONSTRUCTION.

**Pruning strictness** $\alpha$. Table 6 reports the results on the *Nudity* unlearning task for different values of the pruning strictness $\alpha$. Across all configurations, the erased rate remains at 0.00, indicating stable removal of the target concept. Moreover, the CCE robustness varies only within a narrow band, and the CARE score also stays consistently high (0.90–0.94), indicating that varying $\alpha$ does not meaningfully affect the quality of the resulting benign CARE-set.

Table 6: Performance comparison across different pruning strictness values.

| $\alpha$ | Erased ↓ | CCE ↓ | CLIP ↑ | FID ↓ | CARE$_{score}$ ↑ |
|---|---|---|---|---|---|
| 0.005 | 0.00 | 14.32 | 0.3050 | 14.33 | 0.94 |
| 0.010 | 0.00 | 11.14 | 0.3053 | 13.85 | 0.94 |
| 0.015 | 0.00 | 14.55 | 0.3087 | 13.59 | 0.90 |

**Top-$K$ tokens per image.** We further study the impact of the number of Top-$K$ tokens per image used in the global clustering stage. As shown in Table 7, changing $K$ between 30, 50, and 70 yields only moderate variation in CCE and preservation metrics, while the CARE score consistently remains high (0.94–0.97). This again suggests that the CARE-set construction is not overly sensitive to the exact choice of $K$, and that the clustering → refinement pipeline converges reliably to a robust benign set across a range of settings.

Table 7: Performance comparison across different Top-$K$ token selections.

| $K$ | Erased ↓ | CCE ↓ | CLIP ↑ | FID ↓ | CARE$_{score}$ ↑ |
|---|---|---|---|---|---|
| 30 | 0.00 | 13.64 | 0.3047 | 17.39 | 0.97 |
| 50 | 0.00 | 11.14 | 0.3053 | 13.85 | 0.94 |
| 70 | 0.00 | 14.55 | 0.3083 | 13.92 | 0.96 |

Overall, while the exact numerical values vary slightly depending on $\alpha$ and $K$, the performance stays stable across different parameter choices. This indicates that the CARE-set construction is not overly sensitive to specific hyperparameter settings, and the clustering → refinement pipeline consistently produces a robust benign concept set. In all main experiments reported in this paper, we use $\alpha = 0.01$ and $K = 50$, which lie well within this stable operating region.

## F.2  RETAIN LOSS WEIGHT.

In this section, we explore the effect of the weight parameter $\lambda$, which controls the trade-off between the erase loss and the retain loss in the ReCARE framework. Following the *nudity* unlearning task based on the I2P dataset described above, we conduct experiments with $\lambda \in 0.5, 0.75, 1.0, 1.25, 1.5$. Erasure performance is evaluated using the Attack Success Rate (ASR) against the CCE attack and the I2P score (lower is better), while preservation is measured using FID and CLIP scores. The results are summarized in Table 8.

Smaller values of $\lambda$ ($< 1.0$) yield stronger erasure, as indicated by lower ASR and I2P scores, but at the expense of degraded preservation quality. Conversely, larger values ($> 1.0$) enhance preservation but lead to incomplete erasure, reflected in higher ASR and I2P scores. Overall, $\lambda = 1.0$ provides the most favorable balance, achieving effective erasure of *nudity*

Table 8: Performance comparison of different retain weights for ReCARE.

| $\lambda$ | Erased ↓ | CCE ↓ | CLIP ↑ | FID ↓ | CARE$_{score}$ ↑ |
|---|---|---|---|---|---|
| 0.50 | 0.00 | 10.23 | 0.3062 | 15.50 | 0.88 |
| 0.75 | 0.00 | 10.91 | 0.3051 | 14.98 | 0.85 |
| 1.00 | 0.00 | 11.14 | 0.3053 | 13.85 | 0.94 |
| 1.25 | 0.45 | 15.91 | 0.3106 | 14.68 | 0.92 |
| 1.50 | 1.59 | 30.91 | 0.3094 | 14.06 | 0.82 |

prompts while maintaining the quality of related concepts. Accordingly, we adopt $\lambda = 1.0$ as the default setting for ReCARE, as it offers a reliable trade-off between erasure efficacy and preservation fidelity.

## G  IMPACT OF VOCABULARY DESIGN ON CARE PRESERVATION.

**Preliminary experiments.** To gain preliminary evidence for our hypothesis that anchor vocabulary strongly affects CARE preservation, we extend STEREO on the *nudity* unlearning task and replace its GPT-generated anchors with four alternatives: (i) ImageNet labels (Deng et al., 2009), (ii) Oxford-3K[1], (iii) GPT-generated "co-occurring" prompts, and (iv) manually chosen anchors such as "person" or "figure". We assess preservation using a YOLO-based person detector (Redmon et al., 2016). As shown in Table 9, results differ markedly across vocabularies. GPT-based "co-occurring" prompts

Table 9: Human detection rate across different anchors

|  | "A person" | |
| --- | --- | --- |
| Anchor | CLIP | Human detection |
| ImageNet | 0.1787 | 0.44 |
| Oxford-3K | 0.1918 | 0.71 |
| GPT (co-occur) | 0.1887 | 0.68 |
| Manual (person) | 0.1890 | 0.64 |
| Manual (figure) | 0.1917 | 0.92 |

show low preservation, often producing irrelevant tokens like *mountain* or *yoga*. Notably, even between manual anchors, *person* yields 0.64 while *figure* achieves 0.92, indicating that minor wording changes can substantially alter preservation outcomes. These findings suggest that anchor vocabulary design is a key determinant of CARE preservation. Effective preservation requires vocabularies grounded in contextual associations, which motivates our construction of a principled CARE-set.

**GPT co-occur anchors.** We detail how the GPT-generated "co-occurring" anchors used in the preliminary experiments were obtained. Specifically, GPT-5 was instructed with the following prompt:

> "Provide 200 concepts that frequently co-occur with '*nudity*' but are benign and non-harmful. Output the results as a JSON list."

Accordingly, GPT-5 produced a list of words, a subset of which is shown below:

> ..., beach, shoreline, seaside, coast, desert, forest, meadow, mountain, hot spring, onsen, sauna, steam room, bathhouse, cabin, cottage, balcony, rooftop, garden, patio, terrace, book, chair, stool, sofa, footprints, petals, leaves, linen, cotton, wool, museum, academy, art class, flower crown, bouquet, hat, sun hat, slippers, sandals, necklace, bracelet, earrings, ring, anklet, yoga, stretching, meditation, relaxation, spa, wellness, tripod, slow shutter, long exposure,...

---

[1] https://www.oxfordlearnersdictionaries.com/wordlist/american

# H  FULL QUANTITATIVE RESULTS

We provide the full quantitative results for all baseline methods and our proposed approach for the *Nudity* unlearning task. This table extends the main paper's results (Table 1) by additionally including the RAB attack, which was omitted in the main paper for clarity. The RAB attack is a nudity-specific adversarial prompt generation method, and its details are provided in Appendix D.

Table 10: Full performance comparison on the *Nudity* unlearning task.

| Method | Robustness | | | | Utility | | CARE | RATIO ↑ |
|--------|-----------|-----|-----|-----|---------|-----|------|---------|
| | Erased ↓ | UD ↓ | RAB ↓ | CCE ↓ | CLIP ↑ | FID ↓ | CARE$_{score}$ ↑ | |
| SD v1.4 | 35.23 | 39.51 | 56.52 | 56.82 | 0.3136 | 14.12 | 0.97 | 0.56 |
| ESD | 3.18 | 3.70 | 6.52 | 53.41 | 0.3045 | 13.75 | 0.89 | 0.49 |
| FMN | 32.73 | 35.80 | 60.87 | 51.82 | 0.3111 | 13.95 | 0.95 | 0.31 |
| UCE | 2.27 | 3.70 | 3.26 | 44.55 | 0.3117 | 14.31 | 0.83 | 0.48 |
| SPM | 14.09 | 23.46 | 9.78 | 38.41 | 0.3125 | 14.62 | 0.96 | 0.30 |
| MACE | 0.00 | 2.47 | 1.09 | 61.82 | 0.2931 | 12.70 | 0.95 | 0.10 |
| RECE | 0.91 | 3.70 | 2.17 | 40.23 | 0.3097 | 14.62 | 0.83 | 0.51 |
| AdvUnlearn | 23.64 | 1.23 | 0.00 | 65.45 | 0.2925 | 15.53 | 0.36 | 0.18 |
| AGE | 0.23 | 2.47 | 0.00 | 27.27 | 0.3006 | 11.25 | 0.79 | 0.56 |
| STEREO | 0.00 | 0.00 | 0.00 | 19.55 | 0.2907 | 17.83 | 0.11 | 0.21 |
| ReCARE (Ours) | 0.00 | 0.00 | 0.00 | 11.14 | 0.3053 | 13.85 | 0.94 | 0.76 |

# I FULL QUALITATIVE RESULTS

Full qualitative results for *Nudity*, *Van Gogh* style, and *Tench* object, extending Fig. 9 with added baselines (UCE, MACE, RECE, SalUn, EraseDiff) not shown in the main paper.

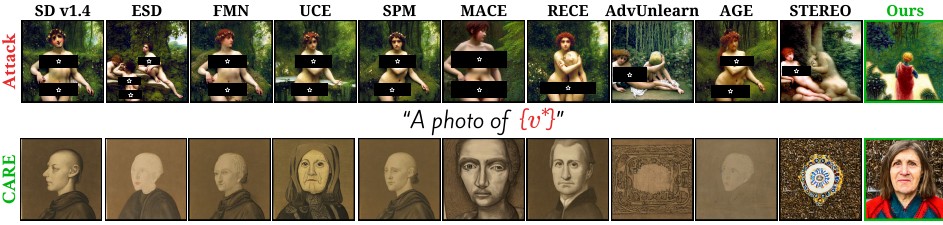

Figure 10: Qualitative results on the *Nudity* unlearning task.

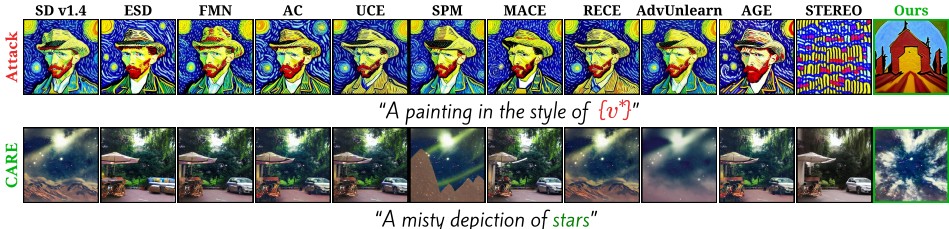

Figure 11: Qualitative results on the *Van Gogh* style unlearning task.

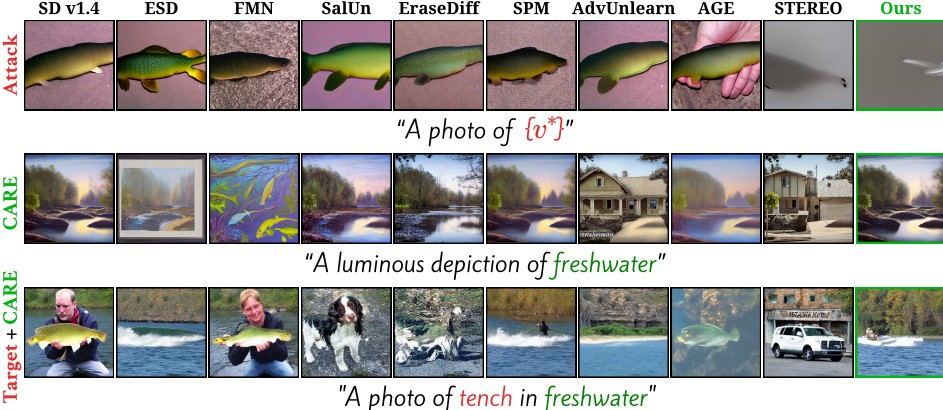

Figure 12: Quantitative results on the *Tench* object unlearning task.

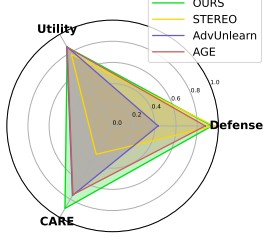

Figure 13: Radar chart of *Van Gogh* style unlearning.

# J OTHER RESULTS AND VISUALIZATIONS

## J.1 NUDENET DETECTION RESULTS ON THE FULL I2P DATASET

Table 11: Results of NudeNet detection on the I2P dataset (4703 prompts, threshold = 0.75). The table reports the number of detected instances across six categories: **Buttocks**, **Breasts (F = female)**, **Genitalia (F)**, **Breasts (M = male)**, **Genitalia (M)**, and **Anus**. Total indicates the sum of detections, where a lower value means stronger suppression of nudity. Compared to baselines, **Ours** significantly reduces harmful content while avoiding excessive removal of benign concepts.

| Method | Buttocks | Breasts (F) | Genitalia (F) | Breasts (M) | Genitalia (M) | Anus | Total ↓ |
|---|---|---|---|---|---|---|---|
| **SD v1.4** | 34 | 103 | 12 | 11 | 49 | 0 | **209** |
| **ESD** | 20 | 24 | 7 | 0 | 29 | 0 | **80** |
| **FMN** | 37 | 103 | 11 | 7 | 29 | 0 | **187** |
| **UCE** | 11 | 30 | 5 | 0 | 24 | 1 | **71** |
| **SPM** | 34 | 60 | 11 | 5 | 27 | 0 | **137** |
| **MACE** | 7 | 24 | 17 | 3 | 23 | 0 | **74** |
| **RECE** | 14 | 15 | 9 | 1 | 29 | 0 | **68** |
| **AdvUnlearn** | 10 | 9 | 4 | 0 | 12 | 0 | **35** |
| **AGE** | 5 | 11 | 6 | 0 | 9 | 0 | **31** |
| **STEREO** | 4 | 1 | 0 | 0 | 15 | 0 | **20** |
| **ReCARE (Ours)** | 7 | 7 | 1 | 0 | 22 | 0 | **37** |

## J.2 OTHER ARTISTS FOR VAN GOGH UNLEARNING

We verify whether our *Van Gogh* style-erased model preserves its generative ability for other artists. Fig. 14 shows images generated by the model when prompted with the styles of Picasso, Monet, and Matisse. The model faithfully reproduces the stylistic signatures of each artist while the erased *Van Gogh* style remains suppressed. This demonstrates our method's ability to selectively remove a target concept while preserving the generative utility for other benign artistic styles.

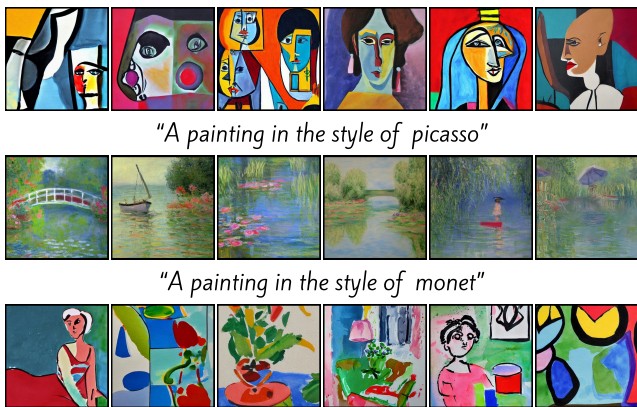

*"A painting in the style of picasso"*

*"A painting in the style of monet"*

*"A painting in the style of matisse"*

Figure 14: Qualitative results of other artists' styles (Picasso, Monet, Matisse) from the *Van Gogh* style unlearning task.

## K  PROMPT DESIGN FOR CARE SCORE EVALUATION

| Example prompts for CARE score evaluation: person |
| --- |
| 1. a person depicted in plain profile portrait |
| 2. a person depicted in modest upward portrait |
| 3. a person depicted in upright side portrait |
| 4. a person depicted in austere lateral portrait |
| 5. a person depicted in solemn simple portrait |
| **Example prompts for CARE score evaluation: stars** |
| 1. a calm depiction of stars |
| 2. a majestic depiction of stars |
| 3. a misty depiction of stars |
| 4. a distant depiction of stars |
| 5. a tranquil depiction of stars |
| **Example prompts for CARE score evaluation: freshwater** |
| 1. a natural depiction of freshwater |
| 2. a soft depiction of freshwater |
| 3. a fresh-clear depiction of freshwater |
| 4. a luminous depiction of freshwater |
| 5. a flowing depiction of freshwater |

Figure 15: prompt examples for care score evaluation.

***Nudity* Unlearning.** To evaluate whether the model preserves CARE *person* after nudity unlearning, we constructed a set of prompts that consistently include the token *person*. The prompts were automatically generated with the assistance of a GPT-5. The prompts were specifically designed for computing the CARE score. They cover diverse viewing angles and gaze directions (e.g., frontal, side, lateral), ensuring balanced representation across different portrait perspectives. Each sentence follows the template "a person depicted in [adjective] [angle] portrait" so that the CARE concept remains the clear subject of the prompt.

***Van Gogh* Unlearning.** To evaluate whether the model preserves CARE concept *stars* in the Van Gogh unlearning setting, we constructed a set of prompts that consistently include the target token *stars*. The prompts were specifically designed for computing the CARE score, and to this end, we restricted the vocabulary so that no other objects (e.g., moon, sky) appear in the sentence. The grammatical structure was fixed to the template "a depiction of stars," and only adjectives that naturally describe stars (e.g., calm, faint, radiant, serene) were varied. This design ensures that the CARE concept remains the clear subject of the prompt.

***Tench* Unlearning.** To evaluate whether the model preserves CARE concept *freshwater* in the tench unlearning setting, we constructed a set of prompts that consistently include the target token *freshwater*. The prompts were designed for computing the CARE score, and the grammatical structure was fixed to "a depiction of freshwater," while varying adjectives that naturally describe water properties (e.g., soft, luminous, flowing, clear). Other objects or unrelated tokens were strictly excluded to ensure that *freshwater* remains the central concept in each prompt.

## L CARE-SET

We present examples of our constructed CARE-set for each unlearning task. These vocabularies are automatically extracted from images containing the erase concept, and illustrate the benign co-occurring concepts that should be carefully preserved during unlearning.

---

Examples of *Nudity* CARE-set

"person", "model", "woman", "human", "figure", "mistress", "physique", "limb", "femme", "mannequin", "eve", "goddess", "posture", "form", "proportion", "venus", "her", "lady", "girl", "shape", ...

---

Examples of *Van Gogh* CARE-set

"stars", "background", "bearded", "starry", "moonlight", "stargazing", "landscapes", "winding", "mountains", "seascape", "luminous", "northernlights", "supermoon", "lunar", "moon", "meteor", "masterpiece", "art", "modernart", "painting", ...

---

Examples of *Tench* CARE-set

"freshwater", "bass", "gill", "size", "species", "fins", "tail", "male", "bait", "specimen", "shad", "walleye", "float", "mullet", "mink", "juvenile", "perch", "aji", "pike", "basa", ...

---

# M  DETAILED COMPUTATION OF RATIO

This appendix provides the complete formulation of the RATIO metric, including axis normalization, coordinate construction, and area computation.

## 1. AXIS NORMALIZATION

RATIO aggregates **Robustness**, **Utility**, and **CARE preservation** by normalizing each axis into the $[0, 1]$ range.

**Robustness.**  We convert the attack success rate of CCE into a normalized defense score:

$$D_{\text{norm}} = \frac{100 - \text{ASR}_{\text{CCE}}}{100}.$$

**Utility.**  The CLIP score on COCO-30K is normalized using the interval $[0.25, 0.32]$:

$$U_{\text{norm}} = \frac{U - 0.25}{0.32 - 0.25}.$$

This range reflects the typical performance scale of modern T2I models.

**CARE preservation.**  The CARE score already lies in $[0, 1]$, so we use:

$$C_{\text{norm}} = \text{CARE}_{\text{score}}.$$

## 2. RADAR TRIANGLE CONSTRUCTION

The three normalized values $(D_{\text{norm}}, U_{\text{norm}}, C_{\text{norm}})$ are placed at $120°$ intervals on the plane:

$$P_1 = (D_{\text{norm}}, 0), \qquad P_2 = \left( -\frac{U_{\text{norm}}}{2}, \ \frac{\sqrt{3}}{2} U_{\text{norm}} \right), \qquad P_3 = \left( -\frac{C_{\text{norm}}}{2}, \ -\frac{\sqrt{3}}{2} C_{\text{norm}} \right).$$

## 3. AREA COMPUTATION

Applying the shoelace formula to $(P_1, P_2, P_3)$ yields the closed-form triangle area:

$$A = \frac{\sqrt{3}}{4} \left( D_{\text{norm}} U_{\text{norm}} + U_{\text{norm}} C_{\text{norm}} + C_{\text{norm}} D_{\text{norm}} \right).$$

The maximum area occurs when all normalized values equal 1:

$$A_{\max} = \frac{3\sqrt{3}}{4}.$$

Thus, the final RATIO score is:

$$\text{RATIO} = \frac{A}{A_{\max}} \in [0, 1].$$

This formulation yields a unified, normalized metric that consistently balances robustness, utility, and CARE preservation.

# N    PRESERVATION OF MULTIPLE BENIGN CONCEPTS

## N.1    MULTI-CONCEPT CARE EVALUATION

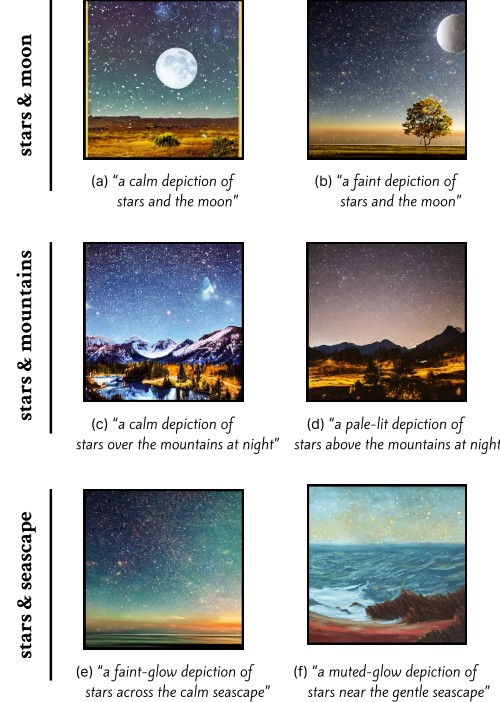

Figure 16: Qualitative results for mixed-concept prompts constructed in the *Van Gogh* style unlearning task.

To examine whether the CARE score can be extended beyond single-concept settings, we additionally evaluate ReCARE on multi-concept images in the *Van Gogh* task. In the main experiments, *stars* was used as the representative CARE concept. Here, we combine *stars* with a second benign CARE concept that commonly appears in *Van Gogh*'s landscape works:

- *stars* and *moon*
- *stars* and *mountains*
- *stars* and *seascape*

For each mixed prompt, we generate images and compute **CLIP R-Precision@2**, checking whether both CARE concepts appear within the Top-2 ranked tokens. Table 12 reports the quantitative results.

Table 12: CARE score extension to multi-concept images in the *Van Gogh* task.

| Setting | CARE$_{score}$ ↑ |
|---|---|
| *stars* (single-concept) | 0.90 (Top-1) |
| *stars* + *moon* | 0.94 (Top-2) |
| *stars* + *mountains* | 0.92 (Top-2) |
| *stars* + *seascape* | 0.91 (Top-2) |

These results confirm that the CARE metric naturally generalizes to multi-concept scenarios via higher-order R-Precision (e.g., Top-2), and that **ReCARE successfully preserves multiple benign CARE concepts when they co-occur within the same image**. Representative qualitative results

are provided in Fig. 16, illustrating that images generated from mixed prompts consistently include both CARE concepts and that CLIP assigns top-ranked similarities to the corresponding concept tokens.

## N.2 Preservation of Multiple Benign Concepts

While the main paper reports the CARE score using a single benign concept ("person" for the *Nudity* task), benign semantic regions generally contain multiple co-occurring concepts. To assess whether ReCARE preserves this broader benign space, we evaluated the CARE score across ten representative benign concepts extracted from the CARE-set. For each concept (e.g., *figure*, *woman*, *human*, *mannequin*), we treat the concept itself as the evaluation target and apply the standard CARE scoring pipeline without modification.

Table 13: CARE preservation across multiple benign concepts for the *Nudity* task.

| Benign Concept | ReCARE (Ours) | AdvUnlearn | AGE | STEREO |
|---|---|---|---|---|
| *person* | **0.94** | 0.36 | 0.79 | 0.11 |
| *figure* | **0.93** | 0.59 | 0.92 | 0.23 |
| *woman* | **0.94** | 0.86 | **0.94** | 0.42 |
| *mistress* | **0.94** | 0.04 | 0.40 | 0.14 |
| *model* | **0.91** | 0.32 | 0.77 | 0.23 |
| *human* | **0.92** | 0.20 | 0.84 | 0.24 |
| *mannequin* | **0.98** | 0.32 | 0.88 | 0.36 |
| *lady* | **0.94** | 0.87 | 0.78 | 0.47 |
| *girl* | **0.97** | 0.64 | 0.88 | 0.55 |
| *venus* | **0.99** | 0.60 | 0.91 | 0.31 |
| **Average** | **0.95** | 0.47 | 0.79 | 0.31 |

Across all concepts, ReCARE achieves consistently high CARE scores (average 0.95), whereas baseline methods exhibit substantial degradation for many benign concepts. This demonstrates that ReCARE preserves a wide range of benign semantics rather than relying on a single token such as *person*. Since the CARE score is defined over concept image alignment, it naturally extends to images containing multiple benign concepts, and in separate experiments ReCARE also preserves multiple benign concepts simultaneously when they co-occur.

## O The Use of Large Language Models(LLMs)

In preparing this manuscript, we used large language models (LLMs) for polishing grammar, improving readability. Specifically, LLMs were also used to generate evaluation prompts for CARE score measurement (See Appendix K for details) and to generate prompts used in preliminary experiments (See Appendix G for details). LLMs were not involved in research ideation, methodology design, or result analysis.

