# OpenReview forum: "Co-occurring Associated REtained concepts in Diffusion Unlearning"
_ICLR.cc/2026/Conference — ICLR 2026 Poster_

### Official Review · Reviewer_GzEp · 2025-10-26

**Soundness:** 2
**Presentation:** 3
**Contribution:** 3
**Rating:** 6
**Confidence:** 3

**Summary:**

This paper addresses the overlooked problem of erasure of benign concepts in unlearning for text-to-image diffusion models. The authors define a CARE score metric to measure this problem and presents a method to create a CARE set to unlearn concepts while retaining the model's ability to generate benign related concepts.

**Strengths:**

- Problem formulation of CARE is practical and highly relevant to the unlearning community. While there have been numerous advances in unlearning methods, quantifying how well benign concepts are retained is crucial.

- CARE score being highly correlated to human ground truths is good evidence that the metric is osund.

- Overall pipeline of filtering for CARE-set and unlearning losses make sense and overall experimental results show good performance over baselines.

- Paper is generally well-written and clear.

**Weaknesses:**

- CARE score tests for one benign concept among 80 unrelated concepts. Most concepts will have multiple benign concepts. Would be ideal to expand to several benign concepts.

- Overall I find the process of creating the CARE-set fairly complicated, for e.g., the clustering steps may be prone to overfitting to things like concepts/prompts and other hyperparameters. Beyond what was presented in appendix F on k-means clusters, have the authors investigated robustness to other aspects and/or are the authors confident the CARE-set construction is robust across parameters?

- Related to above, the training process also seems involved requiring textual inversion for each concept before training. Have the authors quantified the efficiency of their method taking all steps into consideration?

- Insufficient details on how RATIO, the primary evaluation metric, is computed. How is the area normalized?

**Questions:**

- Missing reference [1] which unlearns concepts by substituting with benign alternatives that are manually chosen.

[1] Heng, Alvin, and Harold Soh. "Selective amnesia: A continual learning approach to forgetting in deep generative models." Advances in Neural Information Processing Systems 36 (2023): 17170-17194.

---

> ### Author Response · Authors · 2025-11-21
> **Response by the authors (1/2)**
>
> We appreciate the reviewer’s evaluation of the strengths and weaknesses of our paper. Below, we provide a detailed response addressing each of the weaknesses [W#] and questions [Q#].
>
> ---
> **[W1] Evaluating CARE Preservation with Multiple Benign Concepts**
>
> Thank you for your comment.
>
> We agree that evaluating the CARE score using only a single benign concept can be limiting, since most concepts have multiple benign concepts.
> To address this concern, **we conducted an additional analysis where we used a set of benign co-occurring concepts selected from the CARE-set (10 representative concepts), rather than only the default concept (“*person*”)**.
>
> For the **Nudity task**, we re-computed the CARE score by treating each of these selected benign concepts as the evaluation target, while keeping the evaluation protocol unchanged. The results are shown below:
> | Benign concept | Ours     | AdvUnlearn | AGE      | STEREO   |
> | -------------- | -------- | ---------- | -------- | -------- |
> | person         | **0.94** | 0.36       | 0.79     | 0.11     |
> | figure         | **0.93** | 0.59       | 0.92     | 0.23     |
> | woman          | **0.94** | 0.86       | **0.94** | 0.42     |
> | mistress       | **0.94** | 0.04       | 0.40     | 0.14     |
> | model          | **0.91** | 0.32       | 0.77     | 0.23     |
> | human          | **0.92** | 0.20       | 0.84     | 0.24     |
> | mannequin      | **0.98** | 0.32       | 0.88     | 0.36     |
> | lady           | **0.94** | 0.87       | 0.78     | 0.47     |
> | girl           | **0.97** | 0.64       | 0.88     | 0.55     |
> | venus          | **0.99** | 0.60       | 0.91     | 0.31     |
> | **Average**    | **0.95** | 0.47       | 0.79     | 0.31     |
>
> Across these representative benign concepts, **ReCARE maintains consistently high CARE scores (average 0.95)**, showing that our method does not rely on a particular benign token, but effectively retains a broad set of benign co-occurring concepts.
>
> > If the reviewer’s concern also includes cases where multiple benign concepts appear simultaneously in a single image, we would like to note that we have evaluated this setting separately. As described in our response to Reviewer AnaB (Q2), the CARE score extends naturally to multi-concept images, and ReCARE successfully preserves multiple benign concepts when they co-occur.
>
> We sincerely appreciate the reviewer for highlighting this issue, which motivated us to further validate ReCARE's ability to preserve multiple benign concepts, and we have added this analysis to Appendix N. Thank you.
>
> ---
> **[W2] CARE-set Parameter Sensitivity**
>
> Thank you for raising an important point regarding the parameter sensitivity of the CARE-set construction procedure.
> We conducted additional robustness analyses focusing on the two key parameters described in Appendix C: **pruning strictness (α)** and **Top-K tokens per image (K)**.
>
> **(1) Effect of Pruning Strictness (α)**
> | α              | Erased ↓ | CCE ↓ | CLIP ↑ | FID ↓ | CAREscore |
> | -------------- | -------- | ----- | ------ | ----- | ---------- |
> | 0.005          | 0.00     | 14.32 | 0.3050 | 14.33 | **0.94**   |
> | 0.010          | 0.00     | 11.14 | 0.3053 | 13.85 | **0.94**   |
> | 0.015          | 0.00     | 14.55 | 0.3087 | 13.59 | **0.90**   |
>
> **(2) Effect of Top-K Tokens per Image (K)**
> | K           | Erased ↓ | CCE ↓ | CLIP ↑ | FID ↓ | CAREscore |
> | ----------- | -------- | ----- | ------ | ----- | ---------- |
> | 30          | 0.00     | 13.64 | 0.3047 | 17.39 | **0.97**   |
> | 50          | 0.00     | 11.14 | 0.3053 | 13.85 | **0.94**   |
> | 70          | 0.00     | 14.55 | 0.3083 | 13.92 | **0.96**   |
>
>
> Across all configurations:
>
> - **Erased = 0 consistently**, indicating stable concept removal.
>
> - **CCE remains strong and within a narrow range, preserving SOTA-level robustness.**
>
> - **CARE score remains consistently high (0.90–0.97) regardless of parameter choice.**
>
> Overall, while the exact numerical values vary slightly depending on α and K, **the performance stays stable across different parameter choices.**
> This indicates that the CARE-set construction is **not overly sensitive to specific hyperparameter settings**, and the **clustering → refinement pipeline reliably converges to a robust benign set**.
>
> In all main experiments reported in the paper, we use α = 0.01 and K = 50, which lie well within this stable operating region.
> We have included these sensitivity results in Appendix F. We thank the reviewer again for this helpful feedback.
>
> ---

---

> > ### Author Response · Authors · 2025-11-21
> > **Response by the authors (2/2)**
> >
> > **[W3] End-to-End Efficiency of ReCARE Including Textual Inversion**
> >
> > Thank you for your comment.
> > Although the ReCARE pipeline involves both Textual Inversion and the subsequent unlearning stage, we have quantified the end-to-end cost by measuring all steps involved in the process.
> >
> >  **(1) CARE-set construction cost**.\
> > The entire pipeline (CLIP similarity → clustering → refinement) requires only **1.78 minutes** end-to-end.
> >
> > **(2) Unlearning cost including Textual Inversion**.\
> > ReCARE consists of two components:\
> > (i) **Textual Inversion**, which takes **23.23 minutes**, and\
> > (ii) **ReCARE training**, which takes **5.10 minutes**.
> >
> > The total end-to-end unlearning time is therefore **30.11 minutes**.\
> > A comparison of training time and Nudity task performance with baselines is shown below:
> > | Method            | Time (h) ↓ | CCE ↓ | CLIP ↑ | CAREscore ↑ |
> > | ----------------- | ---------: | ----: | -----: | ----------: |
> > | **ESD**           |       0.69 | 53.41 | 0.3045 |        0.89 |
> > | **RECE (Training-free)**     |       0.01 | 40.23 | 0.3097 |        0.83 |
> > | **AGE**           |       2.20 | 27.27 | 0.3006 |        0.56 |
> > | **AdvUnlearn**    |      21.80 | 65.45 | 0.2925 |        0.36 |
> > | **STEREO**        |       0.41 | 19.55 | 0.2907 |        0.11 |
> > | **ReCARE (Ours)** |       0.50 | 11.14 | 0.3053 |        0.94 |
> >
> > **(3) GPU memory**.\
> > Peak memory consumption during training is approximately **24GB on a single H100 GPU**.
> >
> > These measurements show that, even when accounting for all components including Textual Inversion, the complete ReCARE pipeline finishes in around **30 minutes**, making it efficient and scalable relative to existing unlearning methods.\
> > We have included this computational cost analysis in the EXPERIMENT RESULTS subsection of the revised manuscript. We thank the reviewer for highlighting this important point.
> >
> > ---
> > **[W4] How RATIO is computed and how the area is normalized**
> >
> > Thank you for your comment.
> > We provide additional clarification on how **RATIO** is computed and how the area is normalized.
> >
> > RATIO aggregates **Robustness, Utility**, and **CARE preservation** by first normalizing each axis to the $[0,1]$ range and then computing the area of the resulting radar chart.
> >
> > **(1) Axis normalization**
> > - **Robustness**
> >
> >   We use the attack success rate of **CCE** (Circumventing Concept Erasure) and convert it into a normalized defense score:
> >
> >   $D_{\text{norm}}=\frac{100-\text{ASR}_{\text{CCE}}}{100}$.
> >
> > - **Utility**
> >
> >   The CLIP score on COCO-30K is normalized using the fixed range $[0.25,0.32]$:
> >
> >   $U_{\text{norm}}=\frac{U-0.25}{0.32-0.25}$.
> >
> >   This interval corresponds to the typical performance range of modern T2I models (Stable Diffusion v1.4 scores around 0.31 on this benchmark).
> >
> > - **CARE preservation**
> >
> >   The CARE score already lies in $[0,1]$, so we use:
> >
> >   $C_{\text{norm}}={\text{CARE}_{score}}$.
> >
> > **(2) RATIO**
> >
> > To construct the RATIO metric, we first normalize the three axes and place the corresponding points at 120° intervals.
> >
> > $P_1 = (D_{\text{norm}}\, 0),\quad
> > P_2 = \left(-\frac{U_{\text{norm}}}{2}\, \frac{\sqrt{3}}{2}U_{\text{norm}}\right),\quad
> > P_3 = \left(-\frac{C_{\text{norm}}}{2}\, -\frac{\sqrt{3}}{2}C_{\text{norm}}\right)$.
> >
> > Applying the **shoelace formula** to these three vertices yields the closed-form triangle area:
> >
> > $A = \frac{\sqrt{3}}{4}\left(D_{\text{norm}} U_{\text{norm}}+ U_{\text{norm}} C_{\text{norm}}+ C_{\text{norm}} D_{\text{norm}}\right)$.
> >
> > Since the maximum possible area (when all three normalized values equal 1) is:
> >
> > $A_{\max}=\frac{3\sqrt{3}}{4}$.
> >
> > the final RATIO metric is defined as:
> >
> > $\text{RATIO}=\frac{A}{A_{\max}} \in [0,1]$.
> >
> > This yields a unified and normalized score that balances robustness, utility, and CARE preservation in a consistent and comparable manner.
> >
> > We have included this explanation in Appendix M and will further clarify it in the final version of the paper.
> > We sincerely appreciate the reviewer’s constructive feedback.
> >
> > ---
> > **[Q1] Missing reference: Selective Amnesia (NeurIPS 2023)**
> >
> > Thank you for pointing this out. Selective Amnesia (Heng & Soh, NeurIPS 2023) is relevant prior work on concept forgetting in generative models.
> >
> > We have added this reference to our Appendix E (Related Work) and clarified its relation to our method in the revised version.
> >
> > ---
> > **Overall, we sincerely appreciate your evaluation of our paper, and we are grateful for your comments. Thank you.**

---

### Official Review · Reviewer_AnaB · 2025-10-29

**Soundness:** 3
**Presentation:** 3
**Contribution:** 3
**Rating:** 8
**Confidence:** 3

**Summary:**

This paper shows that, existing machine unlearning (MU) methods also remove the benign co-occurring concepts when removing the target concept. Hence, the paper first introduces the CARE (Co-occurring Associated REtained concepts) score to measure the retention of CARE concepts, and then proposes a method ReCARE (Robust erasure for CAFE) to preserve the model's ability to generate CARE concepts.

Specifically, the CARE score is computed based on CLIP similarity between the generated images conditioned on the prompts containing the chosen CARE concept and the corresponding tokens. Then, the proposed method ReCARE constructs the CARE set based on CLIP similarity and further leverages refinement to filter out unsuitable concepts, and conducts unlearning with retain loss and erase loss using the constructed CARE set.

The proposed CARE score is validated by comparing it with the human-annotated ground truth. Experiments on nudity removal, style unlearning, and object unlearning demonstrate the effectiveness of the proposed method in terms of the robustness and the preservation of benign co-occurring concepts.

**Strengths:**

- The paper is well-written, and the motivation is clear. Existing unlearning methods often fail to generate benign co-occurring concepts, and few metrics consider such measurements. This paper aims for this aspect and seems interesting.
- The proposed CARE score and unlearning method solve the co-occurring concepts missing issue. The method is simple yet effective.

**Weaknesses:**

- The construction of the CARE set involves CLIP similarity computations, t-SNE projection, and k-means clustering, which might incur expensive computational costs for high-resolution scenarios.
- The CARE score and CARE set rely on the CLIP similarity score, while CLIP itself is sensitive to the templates used, which might cause errors when constructing the CARE set.

**Questions:**

- What is the overhead of the proposed method?
- For the CARE score, it considers the Top-1 match for a single chosen concept. Could this be extended to multi-concept images at once?
- Does the template used for CLIP similarity also impact the results?

---

> ### Author Response · Authors · 2025-11-21
> **Response by the authors (1/2)**
>
> We appreciate the reviewer’s evaluation of the strengths and weaknesses of our paper. Below, we provide a detailed response addressing each of the weaknesses [W#] and questions [Q#], with references [#].
>
> ---
> **[Q1 & W1] Overhead and computational cost of CARE-set construction**
>
> Thank you for your comment. Since Q1 (“What is the overhead of the proposed method?”) and W1 (computational cost of CARE-set construction) refer to the same concern, we provide an integrated and consolidated response here.
>
>  **(1) CARE-set construction cost**.
> The entire pipeline (CLIP similarity → clustering → refinement) requires only **1.78 minutes** end-to-end.
>
> **(2) Unlearning cost**.
> ReCARE consists of (i) Textual Inversion (23.23 min) and (ii) ReCARE training (5.10 min), yielding
> **Unlearning time = 28.33 minutes**.
>
> The total end-to-end unlearning time is therefore **30.11 minutes**.\
> A comparison of training time and Nudity task performance with baselines is shown below:
> | Method            | Time (h) ↓ | CCE ↓ | CLIP ↑ | CAREscore ↑ |
> | ----------------- | ---------: | ----: | -----: | ----------: |
> | **ESD**           |     0.69 | 53.41 | 0.3045 |    0.89 |
> | **RECE (Training-free)**     |     0.01 | 40.23 | 0.3097 |  0.83 |
> | **AGE**         |     2.20 | 27.27 | 0.3006 |   0.56 |
> | **AdvUnlearn**    |    21.80 | 65.45 | 0.2925 |   0.36 |
> | **STEREO**        |     0.41 | 19.55 | 0.2907 |  0.11 |
> | **ReCARE (Ours)** |     0.50 | 11.14 | 0.3053 |    0.94 |
>
> **(3) GPU memory**. Peak memory consumption during training is approximately **24GB on a single H100 GPU**.
>
> Both CARE-set construction and ReCARE unlearning are efficient and scalable, completing in around **30 minutes** while maintaining strong performance.
> We have included this computational cost analysis in the EXPERIMENT RESULTS subsection of the revised manuscript.
> We thank the reviewer for highlighting this important point.
>
> ---
> **[W2 & Q3] Robustness of CARE-set Construction and the CARE Score to Templates and Encoders**
>
> Thank you for raising this point. Since Weakness 2 and Question 3 both concern the potential sensitivity of CARE-set construction and CARE score evaluation to CLIP templates, we address them together below for clarity.
>
> **1. Template sensitivity in CARE-set construction**
>
> CARE-set construction uses a single, fixed template only once when generating the initial 500 images.
> The templates are the standard prompts widely used in prior concept-erasure and T2I studies [1, 2, 3, 4 ,5, 6, 7, 8, 9, 10]:
>
> - Nudity: *“A photo of a nude person”*
> - Van Gogh: *“A painting in the style of Van Gogh”*
> - Tench: *“A photo of a tench”*
>
> Importantly, these prompts follow the canonical templates that have become de-facto standards in recent concept erasure and safety alignment literature,
> ensuring that our CARE-set construction is consistent with and directly comparable to prior work.
>
> We also tested alternative templates such as “A picture of …”, “A scene of …”, and found that **the final CARE-set remained nearly unchanged.**
> This robustness comes from the two-stage refinement process (global clustering → intra-cluster pruning), which makes the final CARE-set insensitive to small prompt variations.
>
>
> **2. Template sensitivity in CARE score evaluation**
>
> **(a) CARE score does not rely on a single template**
>
> To avoid template dependence, **evaluation uses 10 diverse templates generated by an LLM, each evaluated with multiple seeds (100 images per method).**
> These templates vary in phrasing, adjectives, scene composition, and perspective.
>
> **(b) Direct test of encoder dependence: CLIP → SigLIP replacement**
>
> To directly test whether CARE score depends on CLIP, we recomputed all CARE scores using SigLIP [7] as the similarity encoder.
>
> Despite differences in absolute values, the **relative ranking across unlearning methods remained consistent under both encoders.**
>
> | Model          | CCE ↓ | SigLIP ↑  | CLIP ↑ |
> | -------------- | --------- | ------- | ---------- |
> | **SD v1.4**  | 56.82    | 0.47 | 0.97  |
> | **STEREO**   | 19.55    | 0.28 | 0.11  |
> | **ESD**     | 53.41    | 0.23 | 0.89   |
> | **UCE**    | 44.55    | 0.28 | 0.91  |
> | **AdvUnlearn** | 65.45    | 0.8  | 0.36   |
> | **AGE**       | 27.27    | 0.12 | 0.79  |
> | **MACE**     | 61.82    | 0.34 | 0.98  |
> | **RECE**     | 40.23    | 0.24 | 0.96  |
> | **SPM**      | 38.41    | 0.39 | 0.96   |
> | **FMN**     | 51.82    | 0.37 | 0.97   |
> | **ReCARE (Ours)**  | 11.14 | 0.40 | 0.94 |
>
>
> This confirms that CARE score is not tied to a particular encoder or template.
> > For a more detailed analysis on encoder dependence (CLIP → SigLIP), we refer the reviewer to our response to Reviewer ut3g (Weakness 1)
>
> **Across both template variations and encoder replacement, we observed consistent CARE-set composition and stable relative CARE score rankings, demonstrating robustness to both template choice and encoder choice.**
>
> We sincerely appreciate this thoughtful and insightful comment.
>
> ---

---

> ### Author Response · Authors · 2025-11-21
> **Response by the authors (2/2)**
>
> **[Q2] Extension of CARE Score to Multi-Concept Images**
>
> Thank you for this insightful suggestion.
>
> While the CARE score is defined for a single CARE concept using CLIP R-Precision@1, **the metric naturally generalizes to multi-concept settings** because it is based on ranking individual concept tokens.
> To further confirm this, **we conducted an additional experiment on the Van Gogh task where two CARE concepts co-occur within the same image**.
>
> We used stars, the CARE concept employed in our main experiments, and constructed mixed-concept prompts such as:
> - *"stars* and *moon"*
> - *"stars* and *mountains"*
> - *"stars* and *seascape"*
>
> These concepts (stars, moon, mountains, seascape) are benign CARE concepts that commonly appear in Van Gogh’s landscape works, making them appropriate for evaluating whether ReCARE preserves multiple CARE concepts when they co-occur.
>
> We then evaluated whether **both concepts appear within the CLIP Top-2 matches (R-Precision@2)**.
> The results are shown below:
> | setting                | CAREscore (ours) |
> | ---------------------- | ----------------- |
> | stars (single-concept) | **0.90 (Top-1)**  |
> | stars + moon           | **0.94 (Top-2)**  |
> | stars + mountains      | **0.92 (Top-2)**  |
> | stars + seascape       | **0.91 (Top-2)**  |
>
> These results demonstrate that the CARE score can be extended to multi-concept images in a straightforward manner, and that ReCARE preserves all benign CARE concepts even when multiple concepts appear simultaneously.\
> We have also added qualitative examples of these multi-concept generations to Appendix N in the revised manuscript, which illustrate that CLIP consistently ranks both CARE concepts at the top positions.
>
> We appreciate the reviewer for raising this point, as it allowed us to clarify the extensibility of CARE evaluation in multi-concept scenarios.
>
> ---
> **Overall, we sincerely appreciate your evaluation of our paper, and we are grateful for your comments. Thank you.**
>
> ---
> **References**
> - [1] Kumari, Nupur, et al. "Ablating concepts in text-to-image diffusion models." Proceedings of the IEEE/CVF International Conference on Computer Vision. 2023.
> - [2] Fan, Chongyu, et al. "Salun: Empowering machine unlearning via gradient-based weight saliency in both image classification and generation." arXiv preprint arXiv:2310.12508 (2023).
> - [3] Lu, Shilin, et al. "Mace: Mass concept erasure in diffusion models." Proceedings of the IEEE/CVF Conference on Computer Vision and Pattern Recognition. 2024.
> - [4] Zhang, Gong, et al. "Forget-me-not: Learning to forget in text-to-image diffusion models." Proceedings of the IEEE/CVF conference on computer vision and pattern recognition. 2024.
> - [5] Bui, Anh, et al. "Fantastic targets for concept erasure in diffusion models and where to find them." arXiv preprint arXiv:2501.18950 (2025).
> - [6] Srivatsan, Koushik, et al. "Stereo: A two-stage framework for adversarially robust concept erasing from text-to-image diffusion models." Proceedings of the Computer Vision and Pattern Recognition Conference. 2025.
> - [7] Zhai, Xiaohua, et al. "Sigmoid loss for language image pre-training." Proceedings of the IEEE/CVF international conference on computer vision. 2023.
> - [8] Chen, Ruidong, et al. "Trce: Towards reliable malicious concept erasure in text-to-image diffusion models." arXiv preprint arXiv:2503.07389 (2025).
> - [9] Biswas, Shristi Das, Arani Roy, and Kaushik Roy. "CURE: Concept Unlearning via Orthogonal Representation Editing in Diffusion Models." The Thirty-ninth Annual Conference on Neural Information Processing Systems. 2025.
> - [10] Gao, Hongcheng, et al. "Meta-unlearning on diffusion models: Preventing relearning unlearned concepts." Proceedings of the IEEE/CVF International Conference on Computer Vision. 2025.

---

> > ### Comment · Reviewer_AnaB · 2025-11-25
> >
> > Thanks for the rebuttal. My concern is fully addressed.

---

> > > ### Author Response · Authors · 2025-11-26
> > >
> > > Thank you very much for your response. We are glad that our clarifications fully addressed your concerns. We sincerely appreciate your time and effort during the review process.

---

### Official Review · Reviewer_eSY1 · 2025-10-31

**Soundness:** 3
**Presentation:** 3
**Contribution:** 3
**Rating:** 6
**Confidence:** 1

**Summary:**

This work creates a set of retaining concepts (CARE) that co-occur with the target concepts but are benign. Using this set of concepts, the authors develop a new unlearning loss that consists of a retain loss and an erase loss. The authors compare their method with standard baselines for diffusion unlearning and demonstrate state-of-the-art performance in balancing concept erasure, utility, and preservation of related concepts.

**Strengths:**

The paper is well-written, and the construction of CARE is well-motivated and explained. The authors performed extensive experiments to compare their method with various baselines.

**Weaknesses:**

(Not a weakness) I am not very familiar with diffusion unlearning and hence am not able to fairly evaluate the quality of this work. I would recommend that the AC seek opinions from other reviewers.

**Questions:**

One question I have is how well does ReCARE preserve the related (but unharmful) concepts that do not appear in the vocabulary $\mathcal{V}$?

---

> ### Author Response · Authors · 2025-11-21
> **Response by the authors**
>
> We appreciate the reviewer’s evaluation of the strengths and weaknesses of our paper. Below, we provide a detailed response addressing each of the questions [Q#].
>
> ---
> **[Q1] Preservation of Related (but unharmful) Concepts Outside the Vocabulary**
>
> Thank you for your comment.
> As we stated in the paper, **our goal is not limited to preserving the specific CARE vocabulary. ReCARE is designed to maintain the overall benign utility of the model, which includes benign concepts beyond the CARE-set**.
> In that sense, our objective aligns well with the question you raised.\
> We would also like to remark that Reviewer VJSB recognized the contribution of our paper that defining CARE not only addresses an overlooked failure case in unlearning but also enables more nuanced and practical unlearning methods that preserve the overall model’s utility.
>
> Our motivation comes directly from our empirical finding that, **among benign concepts, the co-occurring benign concepts that are strongly related to the erase target (i.e., CARE) tend to be unintentionally suppressed** by existing unlearning methods.
> Based on this observation, our method introduces **a clustering-based two stage construction that carefully extracts and captures the target-related co-occurring benign concepts as effectively as possible, ensuring that key related concepts are not missed**, and uses them as preservation signals during unlearning.
>
> Broader benign preservation is also reflected in the COCO-30K utility results, where stable CLIP/FID scores show that the model maintains both the CARE concepts and benign concepts beyond the vocabulary.
>
> **We are grateful for your comment. Thank you.**

---

### Official Review · Reviewer_ut3g · 2025-11-04

**Soundness:** 3
**Presentation:** 3
**Contribution:** 3
**Rating:** 4
**Confidence:** 4

**Summary:**

This paper introduces ReCARE, a framework to improve concept unlearning in diffusion models—specifically addressing the issue that removing harmful concepts (e.g., “nudity”) often unintentionally erases benign co-occurring concepts (e.g., “person”). Authors propose CARE Score, a metric to quantify how well benign co-occurring concepts are preserved after unlearning.
Experiments on three unlearning tasks (nudity, Van Gogh style, and tench object) show ReCARE outperforms prior methods in robustness (low ASR), utility (FID/CLIP), and CARE preservation.

**Strengths:**

- This work identifies an overlooked issue—collateral forgetting of benign co-occurring concepts—and formalizes it as CARE.
- The proposed CARE score offers a measurable dimension beyond robustness and utility.

**Weaknesses:**

- Both CARE-set extraction and CARE score rely heavily on CLIP similarity, which may inherit CLIP’s biases and limit generalization to non-CLIP diffusion models.
- Only three concept domains (nudity, style, object) are tested; it remains unclear how well ReCARE scales to broader or abstract targets (e.g., emotion, violence, or political bias).

**Questions:**

See weaknesses above.

---

> ### Author Response · Authors · 2025-11-21
> **Response by the authors (1/4)**
>
> We appreciate the reviewer’s evaluation of the strengths and weaknesses of our paper. Below, we provide a detailed response addressing each of the weaknesses [W#] and questions [Q#], with references [#].
>
> ---
> **[W1] On the CLIP Dependency of CARE**
>
> Thank you for your comment.
> While our initial use of CLIP follows common practice in diffusion unlearning and safety research [1, 2, 3, 4],
> we understand why this may raise concerns about potential encoder biases.
>
> To examine this, **we fully replaced CLIP with SigLIP** [5] and **CARE showed stable and encoder-agnostic behavior in practice.**\
> The resulting CARE scores are shown below:
> | Model          | CCE ↓ | SigLIP ↑  | CLIP ↑ |
> | -------------- | --------- | ------- | ---------- |
> | **SD v1.4**    | 56.82     | 0.47 | 0.97     |
> | **STEREO**     | 19.55     | 0.28 | 0.11     |
> | **ESD**        | 53.41     | 0.23 | 0.89     |
> | **UCE**        | 44.55     | 0.28 | 0.91     |
> | **AdvUnlearn** | 65.45     | 0.8  | 0.36     |
> | **AGE**        | 27.27     | 0.12 | 0.79     |
> | **MACE**       | 61.82     | 0.34 | 0.98     |
> | **RECE**       | 40.23     | 0.24 | 0.96     |
> | **SPM**        | 38.41     | 0.39 | 0.96     |
> | **FMN**        | 51.82     | 0.37 | 0.97     |
> | **ReCARE (Ours)**       | 11.14     | 0.40 | 0.94 |
>
> Although the absolute values differ due to the representational differences between CLIP and SigLIP, **the relative performance ordering across unlearning methods remains consistent.**
> Methods with strong benign retention under CLIP (e.g., SD v1.4, Ours) also rank highly under SigLIP, while methods with weak retention under CLIP (e.g., AdvUnlearn) remain at the bottom.\
> **This consistency shows that CARE score is not tied to a specific encoder**, and remains robust stable across backbones.
>
> Furthermore, **both the CARE-set and the CARE score are structurally independent of the diffusion model, whether it is CLIP-based or not**.
>
> - **CARE-set construction is performed *before* unlearning and relies solely on an external encoder (e.g., CLIP or SigLIP)**.
>   This step does not rely on the diffusion model’s internal encoder in any form.
> - **CARE score evaluation is performed *after* unlearning by assessing generated images with an external encoder.**
>   The diffusion model is treated as a black box, and only its output images are used to assess whether benign, co-occurring concepts are preserved.
>
> Because of this design, **CARE-set and CARE score are agnostic to the internal encoder of the diffusion model and can be applied consistently to non-CLIP diffusion models without modification.**
>
> We appreciate the concern, which helped us verify CARE’s robustness across encoders.
>
> ---

---

> > ### Author Response · Authors · 2025-11-21
> > **Response by the authors (2/4)**
> >
> > **[W2] On the Choice of Evaluation Domains**
> >
> > Thank you for your comment. However, our evaluation domains (nudity, style, object) correspond to the primary benchmarks used throughout recent diffusion-unlearning research.
> > We surveyed top-tier conferences (CVPR, NeurIPS, ICML, ICCV, ECCV, WACV, AAAI, ICLR) from 2023–2025, and **nearly all methods evaluate on exactly these three categories**.\
> > Thus, our evaluation setup simply follows the common practice established in prior diffusion-unlearning work.
> >
> > | Methods | Nudity | Artistic Style | Object | Violence |
> > |--------|--------|----------------|--------|----------|
> > | EraseDiff (CVPR 2025) [6] | O | X | O | X |
> > | AGE (ICLR 2025) [3] | O | O | O | X |
> > | STEREO (CVPR 2025) [4] | O | O | O | X |
> > | DoCo (AAAI 2025) [7] | X | O | O | X |
> > | Meta-Unlearning (ICCV 2025) [8] | O | O | O | X |
> > | SFD (ICLR 2025) [9] | O | X | X | X |
> > | CoGFD (ICLR 2025) [10] | X | O | O | X |
> > | TRCE (ICCV 2025) [11] | O | O | O | O |
> > | FADE (CVPR 2025) [12] | O | X | O | X |
> > | ACE (CVPR 2025) [13] | O | O | X | X |
> > | GLoCE (CVPR 2025) [14] | O | O | O | X |
> > | EraseFlow (NeurIPS 2025) [15] | O | O | O | X |
> > | SPEED (NeurIPS 2025) [16] | O | O | O | X |
> > | CURE (NeurIPS 2025) [17] | O | O | O | O |
> > | CPE (ICLR 2025) [18] | O | O | O | X |
> > | TIU (CVPR 2025) [19] | O | O | O | X |
> > | AdaVD (CVPR 2025) [20] | O | O | O | X |
> > | EUPMU (NeurIPS 2025) [21] | O | O | O | X |
> > | Semantic Surgery (NeurIPS 2025) [22] | O | O | O | X |
> > | SEMU (ICML 2025) [23] | O | X | O | X |
> > | EraseAnything (ICML 2025) [24] | O | O | O | X |
> > | Co-Erasing (ICML 2025) [25] | O | O | O | X |
> > | LUR (ICCV 2025) [26] | O | X | O | X |
> > | MUNBa (ICCV 2025) [27] | O | X | O | X |
> > | WATER4MU (ICCV 2025) [28] | O | O | O | X |
> > | SuMa (ICCV 2025) [29] | O | O | O | X |
> > | DuMo (AAAI 2025) [30] | O | O | O | X |
> > | SalUn (ICLR 2024) [31] | O | X | O | X |
> > | FMN (CVPR 2024) [32] | O | O | O | X |
> > | SPM (CVPR 2024) [33] | O | O | O | X |
> > | UCE (WACV 2024) [34] | O | O | O | O |
> > | MACE (CVPR 2024) [35] | O | O | O | X |
> > | RACE (ECCV 2024) [36] | O | O | O | O |
> > | RECE (ECCV 2024) [37] | O | O | O | O |
> > | AdvUnlearn (NeurIPS 2024) [2] | O | O | O | X |
> > | ScissorHands (ECCV 2024) [38] | O | X | X | X |
> > | DUO (NeurIPS 2024) [39] | O | X | X | O |
> > | SAeUron (ICML 2024) [40] | O | O | O | O |
> > | Receler (ECCV 2024) [41] | O | O | O | O |
> > | All But One (AAAI 2024) [42] | O | O | O | X |
> > | RGD (NeurIPS 2024) [43] | O | O | O | X |
> > | EAP (NeurIPS 2024) [44] | O | O | O | X |
> > | SFR-on (NeurIPS 2024) [45] | O | X | X | X |
> > | ESD (ICCV 2023) [1] | O | O | O | O |
> > | SA (NeurIPS 2023) [46] | O | X | X | X |
> > | CA (ICCV 2023) [47] | X | O | O | X |
> > | SDD (ICML 2023) [48] | O | O | X | X |
> > | **Total** | **44** | **36** | **40** | **9** |
> >
> > In this regard, broader domains such as **emotion or political bias have not been addressed in prior diffusion unlearning works** and consequently lack standardized benchmarks.\
> > **Violence** is typically treated as a subset of the I2P (Inappropriate Image Prompts) safety categories, alongside nudity or sexual content, and **is therefore not evaluated as an independent domain in recent work**.\
> > We thank the reviewer for raising this point.
> >
> > ---
> > **Overall, we sincerely appreciate your evaluation of our paper, and we are grateful for your comments. Thank you.**
> >
> > ---
> > **References**
> > - [1] Gandikota, Rohit, et al. "Erasing concepts from diffusion models." Proceedings of the IEEE/CVF international conference on computer vision. 2023.
> > - [2] Zhang, Yimeng, et al. "Defensive unlearning with adversarial training for robust concept erasure in diffusion models." Advances in neural information processing systems 37 (2024): 36748-36776.
> > - [3] Bui, Anh, et al. "Fantastic targets for concept erasure in diffusion models and where to find them." arXiv preprint arXiv:2501.18950 (2025).
> > - [4] Srivatsan, Koushik, et al. "Stereo: A two-stage framework for adversarially robust concept erasing from text-to-image diffusion models." Proceedings of the Computer Vision and Pattern Recognition Conference. 2025.
> > - [5] Zhai, Xiaohua, et al. "Sigmoid loss for language image pre-training." Proceedings of the IEEE/CVF international conference on computer vision. 2023.
> > - [6] Wu, Jing, et al. "Erasing undesirable influence in diffusion models." Proceedings of the Computer Vision and Pattern Recognition Conference. 2025.
> > - [7] Wu, Yongliang, et al. "Unlearning concepts in diffusion model via concept domain correction and concept preserving gradient." Proceedings of the AAAI Conference on Artificial Intelligence. Vol. 39. No. 8. 2025.

---

> > > ### Author Response · Authors · 2025-11-21
> > > **Response by the authors (3/4)**
> > >
> > > - [8] Gao, Hongcheng, et al. "Meta-unlearning on diffusion models: Preventing relearning unlearned concepts." Proceedings of the IEEE/CVF International Conference on Computer Vision. 2025.
> > > - [9] Chen, Tianqi, Shujian Zhang, and Mingyuan Zhou. "Score forgetting distillation: A swift, data-free method for machine unlearning in diffusion models." arXiv preprint arXiv:2409.11219 (2024).
> > > - [10] Yao, Quanming, et al. "Erasing concept combination from text-to-image diffusion model." The Thirteenth International Conference on Learning Representations.
> > > - [11] Chen, Ruidong, et al. "Trce: Towards reliable malicious concept erasure in text-to-image diffusion models." arXiv preprint arXiv:2503.07389 (2025).
> > > - [12] Thakral, Kartik, et al. "Fine-Grained Erasure in Text-to-Image Diffusion-based Foundation Models." Proceedings of the Computer Vision and Pattern Recognition Conference. 2025.
> > > - [13] Wang, Zihao, et al. "Ace: Anti-editing concept erasure in text-to-image models." Proceedings of the IEEE/CVF Conference on Computer Vision and Pattern Recognition. 2025.
> > > - [14] Lee, Byung Hyun, Sungjin Lim, and Se Young Chun. "Localized concept erasure for text-to-image diffusion models using training-free gated low-rank adaptation." Proceedings of the Computer Vision and Pattern Recognition Conference. 2025.
> > > - [15] Kusumba, Abhiram, et al. "EraseFlow: Learning Concept Erasure Policies via GFlowNet-Driven Alignment." arXiv preprint arXiv:2511.00804 (2025).
> > > - [16] Li, Ouxiang, et al. "Speed: Scalable, precise, and efficient concept erasure for diffusion models." arXiv preprint arXiv:2503.07392 (2025).
> > > - [17] Biswas, Shristi Das, Arani Roy, and Kaushik Roy. "CURE: Concept Unlearning via Orthogonal Representation Editing in Diffusion Models." NeurIPS. 2025.
> > > - [18] Lee, Byung Hyun, et al. "Concept pinpoint eraser for text-to-image diffusion models via residual attention gate." arXiv preprint arXiv:2506.22806 (2025).
> > > - [19] George, Naveen, et al. "The Illusion of Unlearning: The Unstable Nature of Machine Unlearning in Text-to-Image Diffusion Models." Proceedings of the Computer Vision and Pattern Recognition Conference. 2025.
> > > - [20] Wang, Yuan, et al. "Precise, fast, and low-cost concept erasure in value space: Orthogonal complement matters." CVPR, 2025.
> > > - [21] Zhou, Shiji, et al. "Efficient Utility-Preserving Machine Unlearning with Implicit Gradient Surgery." arXiv preprint arXiv:2510.22124 (2025).
> > > - [22] Xiong, Lexiang, et al. "Semantic Surgery: Zero-Shot Concept Erasure in Diffusion Models." arXiv preprint arXiv:2510.22851 (2025).
> > > - [23] Sendera, Marcin, et al. "SEMU: Singular Value Decomposition for Efficient Machine Unlearning." arXiv preprint arXiv:2502.07587 (2025).
> > > - [24] Gao, Daiheng, et al. "EraseAnything: Enabling concept erasure in rectified flow transformers." ICML, 2025.
> > > - [25] Li, Feiran, et al. "One Image is Worth a Thousand Words: A Usability Preservable Text-Image Collaborative Erasing Framework." arXiv preprint arXiv:2505.11131 (2025).
> > > - [26] Patel, Gaurav, and Qiang Qiu. "Learning to unlearn while retaining: Combating gradient conflicts in machine unlearning." ICCV, 2025.
> > > - [27] Wu, Jing, and Mehrtash Harandi. "Munba: Machine unlearning via nash bargaining." ICCV, 2025.
> > > - [28] Sun, Yuhao, et al. "Invisible Watermarks, Visible Gains: Steering Machine Unlearning with Bi-Level Watermarking Design." ICCV, 2025.
> > > - [29] Nguyen, Kien, Anh Tran, and Cuong Pham. "SuMa: A Subspace Mapping Approach for Robust and Effective Concept Erasure in Text-to-Image Diffusion Models." ICCV, 2025.
> > > - [30] Han, Feng, et al. "Dumo: Dual encoder modulation network for precise concept erasure." AAAI, 2025.
> > > - [31] Fan, Chongyu, et al. "Salun: Empowering machine unlearning via gradient-based weight saliency in both image classification and generation." arXiv:2310.12508 (2023).
> > > - [32] Zhang, Gong, et al. "Forget-me-not: Learning to forget in text-to-image diffusion models." CVPR, 2024.
> > > - [33] Lyu, Mengyao, et al. "One-dimensional adapter to rule them all: Concepts diffusion models and erasing applications." CVPR, 2024.
> > > - [34] Gandikota, Rohit, et al. "Unified concept editing in diffusion models." WACV, 2024.
> > > - [35] Lu, Shilin, et al. "MACE: Mass concept erasure in diffusion models." CVPR, 2024.
> > > - [36] Kim, Changhoon, Kyle Min, and Yezhou Yang. "RACE: Robust adversarial concept erasure for secure text-to-image diffusion model." ECCV, 2024.
> > > - [37] Gong, Chao, et al. "Reliable and efficient concept erasure of text-to-image diffusion models." ECCV, 2024.
> > > - [38] Wu, Jing, and Mehrtash Harandi. "Scissorhands: Scrub data influence via connection sensitivity in networks." ECCV, 2024.
> > > - [39] Park, Yong-Hyun, et al. "Direct unlearning optimization for robust and safe text-to-image models." NeurIPS, 2024.
> > > - [40] Cywiński, Bartosz, and Kamil Deja. "SAeUron: Interpretable concept unlearning in diffusion models with sparse autoencoders." arXiv:2501.18052 (2025).

---

> > > > ### Author Response · Authors · 2025-11-21
> > > > **Response by the authors (4/4)**
> > > >
> > > > - [41] Huang, Chi-Pin, et al. "Receler: Reliable concept erasing of text-to-image diffusion models via lightweight erasers." ECCV, 2024.
> > > > - [42] Hong, Seunghoo, Juhun Lee, and Simon S. Woo. "All but one: Surgical concept erasing with model preservation in text-to-image diffusion models." AAAI, 2024.
> > > > - [43] Ko, Myeongseob, et al. "Boosting alignment for post-unlearning text-to-image generative models." NeurIPS, 2024.
> > > > - [44] Bui, Anh, et al. "Erasing undesirable concepts in diffusion models with adversarial preservation." arXiv:2410.15618 (2024).
> > > > - [45] Huang, Zhehao, et al. "Unified gradient-based machine unlearning with remain geometry enhancement." NeurIPS, 2024.
> > > > - [46] Heng, Alvin, and Harold Soh. "Selective amnesia: A continual learning approach to forgetting in deep generative models." NeurIPS, 2023.
> > > > - [47] Kumari, Nupur, et al. "Ablating concepts in text-to-image diffusion models." ICCV, 2023.
> > > > - [48] Kim, Sanghyun, et al. "Towards safe self-distillation of internet-scale text-to-image diffusion models." arXiv:2307.05977 (2023).

---

### Official Review · Reviewer_VJSB · 2025-11-08

**Soundness:** 3
**Presentation:** 3
**Contribution:** 2
**Rating:** 4
**Confidence:** 3

**Summary:**

This paper identifies a key problem in diffusion model unlearning: the unintended suppression of benign co-occurring concepts. These are defined as CARE (Co-occurring Associated REtained concepts). The authors introduce the CARE score to quantify preservation and propose ReCARE, a method that constructs a CARE-set vocabulary to safeguard these concepts during training. Experiments on targets such as nudity and Van Gogh style show that ReCARE achieves a good balance across erasure robustness, overall utility, and CARE retention.

**Strengths:**

**S1:** This paper identifies and formally defines a previously overlooked issue in diffusion model unlearning: the unintended suppression of benign, co-occurring concepts (CARE). This concept can help develop nuanced, practical unlearning methods that preserve the overall model's utility.

**S2:** This paper provides a framework for the proposed problem. The introduction of the CARE score provides a quantitative metric for objectively measuring concept preservation. The proposed ReCARE method is a practical solution that constructs a "CARE-set" to protect associated concepts during unlearning.

**S3:** This paper conducts a thorough experimental evaluation by testing ReCARE on various targets and comparing it with state-of-the-art baselines, demonstrating the method's effectiveness and generalizability.

**Weaknesses:**

**W1:** The paper does not consider more complex cases in which the target concept overlaps with the retained concept and there are multiple concepts to erase. Adding discussions on these cases would better demonstrate the generalizability and robustness of the proposed method.

**W2:** This paper lacks a comprehensive analysis of efficiency and scalability issues, e.g., the cost of CARE-set construction and the resource consumption of the unlearning process.

**W3:** The automated construction of the CARE-set can introduce biases or errors inherent in models. The paper lacks a thorough discussion of the risk and evaluates the quality of the CARE-set.

**W4:** Although the experiments include a comprehensive set of baselines, the comparison with existing concept erasure approaches that are not based on unlearning (e.g., those based on filtering, post-processing, etc.) is missing.

**Questions:**

See weaknesses.

---

> ### Author Response · Authors · 2025-11-21
> **Response by the authors (1/3)**
>
> We appreciate the reviewer’s evaluation of the strengths and weaknesses of our paper.
> Below, we provide a detailed response addressing each of the weaknesses [W#] and questions [Q#], with references [#].
>
> ---
> **[W1] Addressing More Complex Unlearning Cases**
>
> **Complex case 1. Overlap Between Target and Retained Concepts**
>
> Thank you for your comment.
> As discussed in the main paper, models such as CLIP often encode co-occurring concepts within partially overlapping regions of the embedding space, which naturally leads to entanglement between the target and benign concepts [1]. **Because of this, overlap cases can indeed occur in practical unlearning scenarios, and ReCARE is designed specifically to handle them robustly**.
>
> When an overlapping term is essentially a variant or paraphrase of the target itself (e.g., bare, topless, naked for nudity), retaining it would conflict with the goal of removing the target concept. ReCARE therefore treats such terms as part of the target’s semantic region and filters them out through our clustering-based refinement.
>
> In contrast, concepts such as person, woman, or figure often co-occur with the target but are semantically distinct and should be preserved. **These benign overlaps separate clusters and are included in the CARE-set, with the retain loss ensuring their representations remain intact.**
>
> **Thus, ReCARE explicitly distinguishes between target-like overlaps that should be removed and benign co-occurring overlaps that must be preserved**.
> If the reviewer had a different type of overlap scenario in mind, we would be grateful for a brief example, which would help us address the question more precisely.
>
> **Complex case 2. Multiple Concepts To Erase.**
>
> Thank you for your comment.
> **Most existing concept-erasure methods primarily operate in single-target settings** [2, 3, 4, 5].
>
> Although multi-target unlearning is not the primary focus of our paper, ReCARE is structurally compatible with such extensions.
> This is because each target concept receives its own CARE-set and erase/retain objective, enabling straightforward sequential application.
> Motivated by the reviewer’s comment, **we also conducted a baseline experiment attempting to erase Nudity and Van Gogh simultaneously**.
>
> | **Single Target: Nudity**| | | |**Single Target: Van Gogh**| | | |
> | ------------------------------ | --------- | ---------- | ---------- | ------------------------------ | --------- | ---------- | ---------- |
> | **Model**| **CCE ↓**| **CLIP ↑**| **CAREscore ↑**| **Model**| **CCE ↓**| **CLIP ↑**| **CAREscore ↑**|
> | **ESD**| 53.41| 0.3045| 0.89| **ESD**| 13.20| 0.3074| 0.77|
> | **FMN**| 51.82| 0.3111| 0.95| **FMN**  | 61.60| 0.3140| 0.85|
> | **UCE**| 44.55| 0.3117| 0.83| **UCE**| 61.80| 0.3140| 0.84|
> | **SPM**| 38.41| 0.3125| 0.96| **SPM**| 54.60| 0.3134| 0.82|
> | **MACE**| 61.82| 0.2931| 0.95| **MACE**| 52.40| 0.2862| 0.05|
> | **RECE**| 40.23| 0.3097| 0.83| **RECE**| 55.80| 0.3137| 0.83|
> | **AdvUnlearn**| 65.45| 0.2925| 0.36| **AdvUnlearn**| 57.00| 0.3106| 0.76|
> | **AGE**| 27.27 | 0.3006| 0.79| **AGE**| 12.40| 0.3100| 0.75|
> | **STEREO**| 19.55| 0.2907| 0.11| **STEREO**| 4.00| 0.3047| 0.31|
> | **ReCARE (Ours)**| 11.14 | 0.3053| 0.94| **ReCARE (Ours)**| 6.00| 0.3101| 0.90|
>
>
>
> | **Multi Target: Nudity + Van Gogh** |   |   |   |    |    |
> | ----------------------------------- | ------------------ | ------------------- | ------------------- | -------------------- | ---------- |
> | **Model**        | **CCE(Nudity) ↓** | **CCE(Van gogh) ↓** | **CAREscore(Nudity) ↑** | **CAREscore(Van gogh) ↑** | **CLIP ↑** |
> | **ReCARE (Ours)**| **33.41**          | **26.40**           | **0.90**            | **0.80**             | **0.3113** |
>
>
>
> To provide a clear numerical comparison, we include the single-target baseline results used in our paper in the tables above.
>
> In our multi-target experiment, the total training steps were simply split evenly between the two target concepts, and no additional optimization was applied to account for interactions between them.
> **Despite this disadvantageous setting, ReCARE still maintained competitive robustness against CCE attacks compared to existing single-target unlearning baselines, and its CARE score also remained high.**
>
> This experiment served as a meaningful first step in assessing the extensibility of ReCARE to multi-target settings, and we consider this direction an important avenue for future work. \
> We sincerely thank the reviewer for highlighting this valuable extension.
>
> ---

---

> > ### Author Response · Authors · 2025-11-21
> > **Response by the authors (2/3)**
> >
> > **[W2] Cost of CARE-set construction and resource consumption of the unlearning process.**
> >
> > Thank you for your comment. We provide a detailed breakdown of the computational cost for both CARE-set construction and the full unlearning process.
> >
> >  **(1) CARE-set construction cost**.\
> > The entire pipeline (CLIP similarity → clustering → refinement) requires only **1.78 minutes** end-to-end.
> >
> > **(2) Unlearning cost**.\
> > ReCARE consists of (i) Textual Inversion (23.23 min) and (ii) ReCARE training (5.10 min), summing to **28.33 minutes**.
> >
> > The total end-to-end unlearning time is therefore **30.11 minutes**.\
> > A comparison of training time and Nudity task performance with baselines is shown below:
> > | Method            | Time (h) ↓ | CCE ↓ | CLIP ↑ | CAREscore ↑ |
> > | ----------------- | ---------: | ----: | -----: | ----------: |
> > | **ESD**           |       0.69 | 53.41 | 0.3045 |        0.89 |
> > | **RECE (Training-free)**     |       0.01 | 40.23 | 0.3097 |        0.83 |
> > | **AGE**           |       2.20 | 27.27 | 0.3006 |        0.56 |
> > | **AdvUnlearn**    |      21.80 | 65.45 | 0.2925 |        0.36 |
> > | **STEREO**        |       0.41 | 19.55 | 0.2907 |        0.11 |
> > | **ReCARE (Ours)** |       0.50 | 11.14 | 0.3053 |        0.94 |
> >
> > **(3) GPU memory**.\
> > Peak memory consumption during training is approximately **24GB on a single H100 GPU**.
> >
> > Both CARE-set construction and ReCARE unlearning are efficient and scalable, completing in around **30 minutes** while maintaining strong performance.\
> > We have included this computational cost analysis in the EXPERIMENT RESULTS subsection of the revised manuscript.
> > We thank the reviewer for highlighting this important point.
> >
> > ---
> > **[W3] Risks of automatic CARE-set construction and its quality evaluation**
> >
> > Thank you for your comment.
> > Existing unlearning methods (e.g., STEREO, AdvUnlearn) typically rely on manually selected anchors or external vocabularies, which often do not reflect the actual concepts that co-occur with the target in real images.
> > This limitation motivated us to automatically construct the CARE-set directly from the target images. As the reviewer noted, however, automatic extraction can include incorrectly selected or noisy tokens, so we additionally verified the quality of the extracted CARE-set.
> >
> > To assess semantic validity, **we applied a GPT-4o-mini binary validator (“yes/no”),
> > following prior LLM-based concept-validation work** [6, 7, 8, 9, 10, 11, 12], which checks whether
> > each extracted token is semantically appropriate as a benign co-occurring concept.
> > The validation template used in this check is provided below.
> >
> > The automatically extracted tokens showed high agreement:
> > | Target   | YES Ratio |
> > | -------- | --------- |
> > | **Nudity**   | 0.914     |
> > | **Van Gogh** | 0.909     |
> > | **Tench**    | 0.949     |
> >
> > **We also examined several actual tokens (person, stars, freshwater) produced by our method using human annotation (Fig. 4)**. These are consistent with benign concepts humans typically expect to be preserved, and prior unlearning methods often fail specifically on these concepts. **This supports that the automatic construction identifies the correct benign co-occurring concepts rather than arbitrary ones.**
> >
> > **For transparency, Appendix L provides representative CARE-set examples for each target**, allowing direct inspection of their semantic quality. Below are several representative examples (taken from Appendix L):
> > - Nudity: *person, woman, figure, human, mannequin, form, shape, lady,* ...
> > - Van Gogh: *stars, moonlight, seascape, supermoon, mountains, landscapes, art,* ...
> > - Tench: *freshwater, gill, fins, walleye, bass, bait, tail, man,* ...
> >
> > >
> > 	You are an expert concept validator.
> >
> > 	A CARE concept is a normal, benign concept that **often appears together with the
> > 	target concept in images**, but it does not represent the target concept itself.
> >
> > 	Your task:
> > 	Given a list of (concept, target_concept) pairs, determine whether each concept
> > 	is an appropriate CARE concept for that target.
> >
> > 	Respond:
> > 	- "yes" if the concept is a normal, benign co-occurring concept.
> > 	- "no" if the concept represents the target concept itself or directly expresses its sensitive meaning.
> >
> > 	Return one line containing the same number of answers as the input pairs,
> > 	using "yes" or "no", separated by commas.
> >
> > Thank you again for your thoughtful comment.
> >
> > ---

---

> ### Author Response · Authors · 2025-11-21
> **Response by the authors (3/3)**
>
> **[W4] Comparing Against Non-Unlearning Approaches**
>
> Thank you for your comment.
> However, **filtering or post-processing defenses are not comparable under our evaluation scheme**.
> As discussed in Appendix E, they do not modify model parameters, are easily bypassed by adversarial or inversion attacks,
> and their effects on utility or CARE are not assessed on the same evaluation axis.
>
> For this reason, **prior diffusion unlearning work benchmarks methods that perform model-level edits, and our evaluation includes the core baselines used in this literature**.
>
> To address the reviewer’s concern, **we additionally evaluated SAFREE [13] as a representative filtering/post-processing method** under the same metrics.
> For context, we also include UCE [14] and RECE [15], which are training-free model-editing approaches that modify cross-attention weights in closed form and represent the closest comparable category within this setting.
>
> The results are summarized below:
> | Method                   | ASR ↓     | CLIP ↑ | CAREscore ↑ | RATIO ↑  |
> | ------------------------ | --------- | ------ | ----------- | -------- |
> | **UCE**  | 44.55     | 0.3117 | 0.83        | 0.48     |
> | **RECE** | 40.23     | 0.3097 | 0.83        | 0.51     |
> | **SAFREE** | 34.11   | 0.3110 | 0.96        |  0.68    |
> | **ReCARE (Ours)**        | 11.14 | 0.3053 | 0.94    | 0.76 |
>
> Across these methods, ReCARE provides a competitive balance among robustness, utility, and CARE preservation.
> We thank the reviewer again for raising this point.
>
> ---
> **Overall, we sincerely appreciate your evaluation of our paper, and we are grateful for your comments. Thank you.**
>
> ---
> - [1] Jiang, Kenan, et al. "Comclip: Training-free compositional image and text matching." Proceedings of the 2024 Conference of the North American Chapter of the Association for Computational Linguistics: Human Language Technologies (Volume 1: Long Papers). 2024.
> - [2] Srivatsan, Koushik, et al. "Stereo: A two-stage framework for adversarially robust concept erasing from text-to-image diffusion models." Proceedings of the Computer Vision and Pattern Recognition Conference. 2025.
> - [3] Bui, Anh, et al. "Fantastic targets for concept erasure in diffusion models and where to find them." arXiv preprint arXiv:2501.18950 (2025).
> - [4] Zhang, Yimeng, et al. "Defensive unlearning with adversarial training for robust concept erasure in diffusion models." Advances in neural information processing systems 37 (2024): 36748-36776.
> - [5] Wu, Jing, et al. "Erasing undesirable influence in diffusion models." Proceedings of the Computer Vision and Pattern Recognition Conference. 2025.
> - [6] Hu, Yushi, et al. "Tifa: Accurate and interpretable text-to-image faithfulness evaluation with question answering." Proceedings of the IEEE/CVF International Conference on Computer Vision. 2023.
> - [7] Ye, Ziyi, et al. "Learning LLM-as-a-judge for preference alignment." The Thirteenth International Conference on Learning Representations. 2025.
> - [8] Mañas, Oscar, Benno Krojer, and Aishwarya Agrawal. "Improving automatic vqa evaluation using large language models." Proceedings of the AAAI Conference on Artificial Intelligence. Vol. 38. No. 5. 2024.
> - [9] Guan, Tianrui, et al. "Hallusionbench: an advanced diagnostic suite for entangled language hallucination and visual illusion in large vision-language models." Proceedings of the IEEE/CVF Conference on Computer Vision and Pattern Recognition. 2024.
> - [10] Li, Minghan, et al. "FiVE-Bench: A Fine-grained Video Editing Benchmark for Evaluating Emerging Diffusion and Rectified Flow Models." Proceedings of the IEEE/CVF International Conference on Computer Vision. 2025.
> - [11] Wu, Tong, et al. "FiVA: Fine-grained visual attribute dataset for text-to-image diffusion models." Advances in Neural Information Processing Systems 37 (2024): 31990-32011.
> - [12] Wu, Tianhao, et al. "Meta-rewarding language models: Self-improving alignment with llm-as-a-meta-judge." Proceedings of the 2025 Conference on Empirical Methods in Natural Language Processing. 2025.
> - [13] Yoon, Jaehong, et al. "Safree: Training-free and adaptive guard for safe text-to-image and video generation." arXiv preprint arXiv:2410.12761 (2024).
> - [14] Gandikota, Rohit, et al. "Unified concept editing in diffusion models." Proceedings of the IEEE/CVF Winter Conference on Applications of Computer Vision. 2024.
> - [15] Gong, Chao, et al. "Reliable and efficient concept erasure of text-to-image diffusion models." European Conference on Computer Vision. Cham: Springer Nature Switzerland, 2024.

---

### Author Response · Authors · 2025-12-02
**Rebuttal Summary for AC**

Dear AC,\
We sincerely thank you for your time and for overseeing the discussion process.\
The reviewers raised thoughtful and valid questions across efficiency, robustness, extensibility, complex unlearning settings, and comparisons with alternative approaches. **Engaging with these points has strengthened the paper and clarified its contributions**.

Below, we summarize how each main concern was addressed across the five reviews.

---
## **Strength**
Across the reviews, several strengths of our work were consistently highlighted:

- Identification and formalization of **CARE (Co-occurring Associated REtained concepts)** as a previously overlooked failure mode in diffusion unlearning (VJSB, ut3g, AnaB, GzEp)
- Introduction of **CARE score**, a general and quantitative metric for benign concept preservation (VJSB, ut3g, AnaB, GzEp)
- **ReCARE**, a practical and effective framework **balancing robust erasure, utility, and CARE preservation** (VJSB, AnaB, GzEp)
- Clear motivation, strong writing, and extensive experimental evaluation (VJSB, eSY1, AnaB, GzEp)

We are grateful to the reviewers for recognizing these contributions.

---
## **Main Concerns Raised & How They Were Addressed**

**1. Efficiency & Resource Cost** (VJSB, AnaB, GzEp)
We reported full **end-to-end cost** (CARE-set → Textual Inversion → ReCARE), peak GPU memory, and comparisons with baselines.
ReCARE completes a full unlearning run in **~30 minutes** while still outperforming heavier baselines.

**2. Robustness of CARE-set and CARE score** (VJSB, ut3g, AnaB, GzEp)
We evaluated robustness across (i) CLIP → SigLIP **encoder variation**, (ii) **hyperparameter sensitivity**, and (iii) **LLM-based semantic validation**.
Across all settings, **CARE-set composition and overall performance remained stable**, showing that CARE is robust to encoder and parameter changes.

**3. CARE score Extension to Multi-Concept and Broader Benign Concepts** (AnaB, GzEp)
We tested CARE score using (i) images where **multiple CARE concepts appear together** and (ii) **several different benign concepts** drawn from the CARE-set.
CARE score stayed consistent in these different situations, showing that the metric naturally **extends beyond the default single-concept case while maintaining its performance**.

**4. Considering More Complex Unlearning Settings** (VJSB)
We clarified how ReCARE handles overlap between the target and benign concepts by CARE design and **additionally evaluated a multi-target case** without modifying the framework.
ReCARE showed **competitive performance**, indicating that multi-target extensions are feasible directions for **future work**.

**5. Comparison with Non-Unlearning Methods** (VJSB)
Although filtering/post-processing methods fall outside the main unlearning axis, **we added SAFREE** for completeness.
ReCARE **still showed a balanced performance** across robustness, utility, and CARE preservation in the comparison.

---
These revisions and clarifications are reflected in the updated manuscript.\
We hope that the revisions will be helpful for your decision and will merit favorable consideration of our submission.

Thank you very much for your time and attention.

Best regards,\
Authors of Submission 16776

---

### Meta-Review · Area_Chair_wJEk · 2026-01-17

**Summary:**

This paper introduces ReCARE, a framework to improve concept unlearning in diffusion models—specifically addressing the issue that removing harmful concepts (e.g., “nudity”) often unintentionally erases benign co-occurring concepts (e.g., “person”).

Reviewers expressed the following concerns.
- Paper addresses simple cases; request adding discussion on complex cases with multiple concept removal.
- Paper lacks analysis of efficiency and scalability
- Proposed method rely on CLIP similarity, which itself can be biased.
- Only three "concept domains (nudity, style, object)" are tested. Generalization of the proposed method beyond these is unclear.
- Generalization of the proposed method to concept that do not appear in the selected vocabulary (COCO labels?)
- Process for creating CARE-set is a bit complicated with some ad hoc steps.
- Missing details about RATIO metric

**Reviewer Concerns:**

Authors provided responses to all reviewer comments.

- Regarding multiple concept erasure; the focused on nudity and style. I feel the reviewers were interested in more general concepts/objects that should be tested to evaluate whether co-occurring benign concepts/objects can be retained.
- Regarding complexity and computational cost, authors provided the details and a comparison with other methods.
- Regarding multiple concept domains, authors argue that prior work use the same domains as tested in this paper.

- Regarding retaining multiple benign concepts, authors agreed with the reviewer concerns and added an experiment by selecting 10 representative concepts. While the results are limited but they highlight the capabilities of the proposed method.

**Reviewer Scores:**

Reviewer scores are 4,4,6,8, 6

One reviewer with 6 rating had confidence level 1 and explicitly stated that "I am not very familiar with diffusion unlearning and hence am not able to fairly evaluate the quality of this work. I would recommend that the AC seek opinions from other reviewers."

Overall, I feel that authors have addressed the reviewer concerns to an acceptable level.
Additional experiments should be included in the final version if the paper is accepted.

---

### Decision · Program_Chairs · 2026-01-26

Accept (Poster)